# Contaminant source localization via Bayesian global optimization

Guillaume Pirot[1], Tipaluck Krityakierne[2,3,4], David Ginsbourger[4,5,6], and Philippe Renard[7]

[1]Institute of Earth Sciences, University of Lausanne, Switzerland
[2]Department of Mathematics, Faculty of Science, Mahidol University, Bangkok, Thailand
[3]Centre of Excellence in Mathematics, CHE, Bangkok, Thailand
[4]Oeschger Center for Climate Change Research, University of Bern, Switzerland
[5]Uncertainty Quantification and Optimal Design group, Idiap Research Institute, Martigny, Switzerland
[6]Institute of Mathematical Statistics and Actuarial Science, University of Bern, Switzerland
[7]Centre for Hydrogeology and Geothermics, University of Neuchâtel, Switzerland

*Correspondence to:* G. Pirot (guillaume.pirot@unil.ch)

**Abstract.** Contaminant source localization problems require efficient and robust methods that can account for geological heterogeneities and accomodate relatively small data sets of noisy observations. As realism commands hi-fidelity simulations, computation costs call for global optimization algorithms under parcimonious evaluation budgets. Bayesian optimization approaches are well-adapted to such settings as they allow exploring parameter spaces in a principled way so as to iteratively locate the point(s) of global optimum while maintaining an approximation of the objective function with an instrumental quantification of prediction uncertainty. Here, we adapt a Bayesian optimization approach to localize a contaminant source in a discretized spatial domain. We thus demonstrate the potential of such method for hydrogeological applications and also provide test cases for the optimization community. The localization problem is illustrated for cases where the geology is assumed to be perfectly known. Two 2D synthetic cases that display sharp hydraulic conductivity contrasts and specific connectivity patterns are investigated. These cases generate highly non-linear objective functions that present multiple local minima. A derivative-free global optimization algorithm relying on a Gaussian Process model and on the Expected Improvement criterion is used to efficiently localize the point of minimum of the objective functions, which corresponds to the contaminant source location. Even though concentration measurements contain a significant level of proportional noise, the algorithm localize efficiently the contaminant source location. The variations of the objective function are essentially driven by the geology, followed by the design of the monitoring well network. The data and scripts used to generate objective functions are shared to favour reproducible research. This contribution is important because the functions present multiple local minima and are inspired from a practical field application. Sharing these complex objective functions provides a source of test cases for global optimization benchmarks and should help designing new and efficient methods to solve this type of problem.

## 1 Introduction

Many hydrogeological processes are governed by nonlinear equations (e.g. unsaturated flow problems, heat and transport problems; De Marsily, 1986). This often results in highly non-linear and non-convex responses in the objective functions of related optimization problems. Often, however, default optimization algorithms employed in the hydrogeological community,

notably concerning contaminant source localization problems, are based on local search principles (using analytical gradients or estimates thereof) (Mahar and Datta, 2000; Ayvaz, 2016). In contrast, derivative-free global optimization methods such as evolutionary algorithms, simulated annealing and others have also become commonplace in the last decades. Yet, these are typically regarded with caution as they do not systematically come with much guarantees, and can potentially require large numbers of function evaluations, a situation that is to be avoided in the case where forward runs are CPU-intensive. On the other hand, so-called Bayesian optimization algorithms have been considerably gaining importance in several fields lately as they enjoy a number of practical advantages while having been recently proven to possess desirable consistency properties (Vazquez and Bect, 2010; Bect et al., 2018). One of the greatest strengths of common Bayesian optimization algorithms is that they do not only guide evaluations towards the global optimum but also maintain an approximate representation of the objective function together with a quantification of prediction uncertainty. This enables space exploration with a memory so as to prevent or mitigate evaluations at redundant locations. In addition, recent adaptations of popular Bayesian optimization approaches allow accommodating evaluation noise (Picheny and Ginsbourger, 2014a), parallel evaluations (Marmin et al., 2015), high dimensions (Wang et al., 2018), non-stationarity (Snoek et al., 2014), gradient observations (Wu et al., 2017), and many more features (see for instance Ginsbourger (To appear) for a broader overview of sequential design algorithms for computer experiments).

Despite the fantastic rise of Bayesian optimization in machine learning and for the design of computer in various communities, its spread in the geosciences remains relatively modest so far, perhaps in part because contrarily to analytical test functions inspired by engineering problems or off-the-shelf machine learning algorithms trained on openly available data bases, it is often the case (e.g., in heavy flow simulations) that geoscientific data and/or computer codes cannot be publicly shared or easily handled for one or the other reason. One way around that is to share instead a finite number of evaluation results performed onsite by the authors, so that users do not need to run new simulations when testing their algorithms. Yet, optimization algorithms typically used in hydrogeological inverse and related problems assume a continuous search space. When relying on a discretization of the input space, possible options that come to mind include i) using discrete optimization algorithms, ii) using continuous optimization algorithms on a re-interpolated function based on available evaluation results, and iii) constraining continuous optimization methods by forcing novel evaluation points to remain among the considered finite set. Our approach here, guided by the willingness to share data with both geosciences and optimization communities and produce reproducible research, is to rely on a fine grid of evaluation results that allows appealing to any of the aforementioned three approaches. By remaining in a two-dimensional framework, our discretized 2601-element data set is actually fine enough to capture the complex behavior of the considered misfit function so that optimization algorithms can be possibly compared by users on high-fidelity approximations of the objective. This being said, we rather insist throughout the paper on a natural adaption of a popular Bayesian optimization algorithm to the discrete case, hence simultaneously addressing points i) and iii) above and thus demonstrating the applicability of this family of techniques both to a challenging contaminant localization problems in general and to discrete situations in particular (that could be relevant in a number of practical situations, e.g. well placement).

Coming back to more specific hydrogeological concerns underlying our application test case, contaminant characterization problems are motivated by the fact that the concept of polluter pays (OECD, 1972) holds for groundwater protection laws in

many countries (USA, 1972; Swiss Confederation, 1983; European Union, 2000). A polluter can sometimes be identified by a specific chemical signature (Mansuy et al., 1997; Rachdawong and Christensen, 1997; Venkatramanan et al., 2016). However, when the signature is not unique, the ability to localize the contaminant source(s) can make defining responsibilities or reducing decontamination costs easier. Moreover, contaminant transport is dominated by the heterogeneity of the subsurface properties.

In particular, it is controlled by the connectivity of geobodies (e.g. characterized by lithofacies or by a range of hydraulic conductivity values) and by the sharpness of geobody property contrasts.

Thus, solving contaminant source localization problems in complex environments characterized by strong property contrasts requires methods that are robust, time efficient and able to handle input data uncertainty. Several approaches have been proposed in the last three decades (Atmadja and Bagtzoglou, 2001; Amirabdollahian and Datta, 2013). Analytical solutions (Ala

and Domenico, 1992; Alapati and Kabala, 2000) are limited to homogeneous geological media. Methods able to handle heterogeneous geological medium can be classified into two groups: backward or forward solver based approaches. The backward solvers consist of reversing the flow problem (Skaggs and Kabala, 1995; Milnes and Perrochet, 2007; Ababou et al., 2010) and solving the Advection Dispersion Equation backward in time to localize the source and identify the release history. In this group of methods, both the flow-field and the contaminant plume are assumed perfectly known. Methods using forward solvers

are based on an inverse problem formulation (Aral et al., 2001; Yeh et al., 2007; Mirghani et al., 2012), where the source location and release history are inferred from concentration samples. Parameter sets are proposed and used as inputs in a forward solver to simulate concentration breakthrough curves at the sample locations; when the mismatch between the simulated concentrations and the observed ones is within an acceptable level of error, the proposed model is accepted as a solution. The best solution is the one minimizing the error function. In this group of methods, less information about the contaminant plume

is required and the method can be adapted to uncertain geology (Zhang et al., 2016). To the best of our knowledge, existing studies using forward solver based approaches were limited to homogeneous (Datta et al., 2011; Hansen and Vesselinov, 2016) or multi-Gaussian like (Aral et al., 2001; Ayvaz, 2016) heterogeneous property field.

Within this last set of methods, different optimization techniques can be employed. Classical non-linear optimization techniques following a gradient based approach (Mahar and Datta, 2000; Datta et al., 2011), such as the Levenberg-Marquardt

algorithm (Hansen and Vesselinov, 2016), present the risk of being stuck in local minima. Employing a tabu search algorithm (Yeh et al., 2007) presents the same inconvenience as it explores iteratively neighbor solutions. Combining a gradient descent algorithm with a genetic algorithm (Aral et al., 2001; Ayvaz, 2016) decreases the risk of becoming stuck in local minima, but the genetic algorithm may require longer parameter exploration if the mutations are not guided by a smart rule. Simulated annealing (Amirabdollahian and Datta, 2014) allows for a broader exploration but at a very high computational cost. Bayesian

optimization[1] is a global approach that limits considerably the risk of being trapped in local minima and does not require the computation of derivatives of the objective function. It explores smartly the parameter space by looking at figures of merit

---

[1]While Bayesian methods have been massively used throughout groundwater sciences and notably for contaminant source localization, let us emphasize that the term 'Bayesian optimization' does not refer to any arbitrary method that combines 'optimization' and 'Bayesian statistics'. Instead, the term refers to a specific family of optimization algorithms where a prior distribution is put on the objective function (See e.g. Shahriari et al., 2016, and references therein for an overview).

such as the Expected Improvement (EI) criterion (Mockus, 1989; Jones et al., 1998a; Vazquez and Bect, 2010), trading off exploitation of available results and space exploration. To the best of our knowledge, the potential of this class of methods for addressing contaminant source localization problems is still unexplored.

The objective of this paper is threefold.

The first objective is to assess the performance of an inverse problem formulation to identify contaminant source characteristics on a synthetic case displaying strong hydrogeological property contrasts and complex connected structures. This is important because in spite of its advantages, inverse problem formulation to identify contaminant source characteristics has been employed only on multi-Gaussian type heterogeneities and the type of heterogeneities strongly influences mass transport.

The second objective is to test the efficiency and advantages of a Bayesian optimization algorithm which relies on expected improvement criteria in the formulated contaminant source identification problem. While Bayesian optimization has been applied to a variety of optimization problems, we believe that this is the first time the algorithm has been applied to the contaminant source identification problem.

And last but not least, the third objective is to provide an open source optimization benchmark case that allows comparing
different optimization strategies on objective functions defined over a discrete domain and inspired by real applications, which are not currently available in the optimization community.

With these objectives, we propose an original application of an EI algorithm to infer, in a deterministic inverse problem formulation, the contaminant source location in a 2D heterogeneous aquifer that presents strong property contrasts and complex connected structures. To allow for a comparison between the optimizer exploration and an exhaustive search of the discrete
parameter space, the model grid is limited to 2D to keep computational cost reasonable for flow and transport simulations. The hydraulic conductivity field is generated with the multiple-point statistics algorithm called *DeeSse* (Straubhaar et al., 2016), from a training image representing the heterogeneous hydrogeological properties of a braided-river aquifer, which was generated by a pseudo-genetic algorithm (see Appendix A; Pirot et al., 2015). The hydrogeological properties and flow boundary conditions are assumed to be perfectly known. The flow and transport equations are solved numerically using the
*Groundwater* software (Cornaton, 2007). Because measurement error and monitoring network affect the objective function, the algorithm performances are tested for different levels of measurement error and for different configurations of monitoring wells. The optimization is performed using the DiceKriging and DiceOptim R packages (Roustant et al., 2012). In addition, we provide a benchmark for optimization algorithms, which relies on an objective function generator that can be customized by choosing between 2 geological scenarios, 2 possible locations for the contaminant source and by the selection of observations
among 25 monitoring wells. The performance of the EI algorithm is assessed by 100 runs from different initial designs.

The paper is organized as follows. Section 2 describes the synthetic test case and the experimental setup. Section 3 explains the objective function generator. Section 4 details the steps of the EI algorithm. The results are presented in Section 5 and are discussed in Section 6. Conclusions are summed up in Section 7. The supplementary material provided online is listed in Appendix.

## 2  Synthetic test cases

As different geological settings can lead to very different objective functions, and in order to test the robustness of the optimization method, we consider two synthetic cases corresponding to 5 m thick × 600 m long × 300 m wide braided river aquifers. Each aquifer is represented by a unique, supposedly known, 2D facies model (Figure 1) of 1 m by 1 m resolution to simplify the problem and to decrease the computing costs related to transport simulations. These 2D facies models (Figure 1), which present strong contrasts and realistic spatial structures, are generated by multiple-point statistics (MPS) simulation, using the

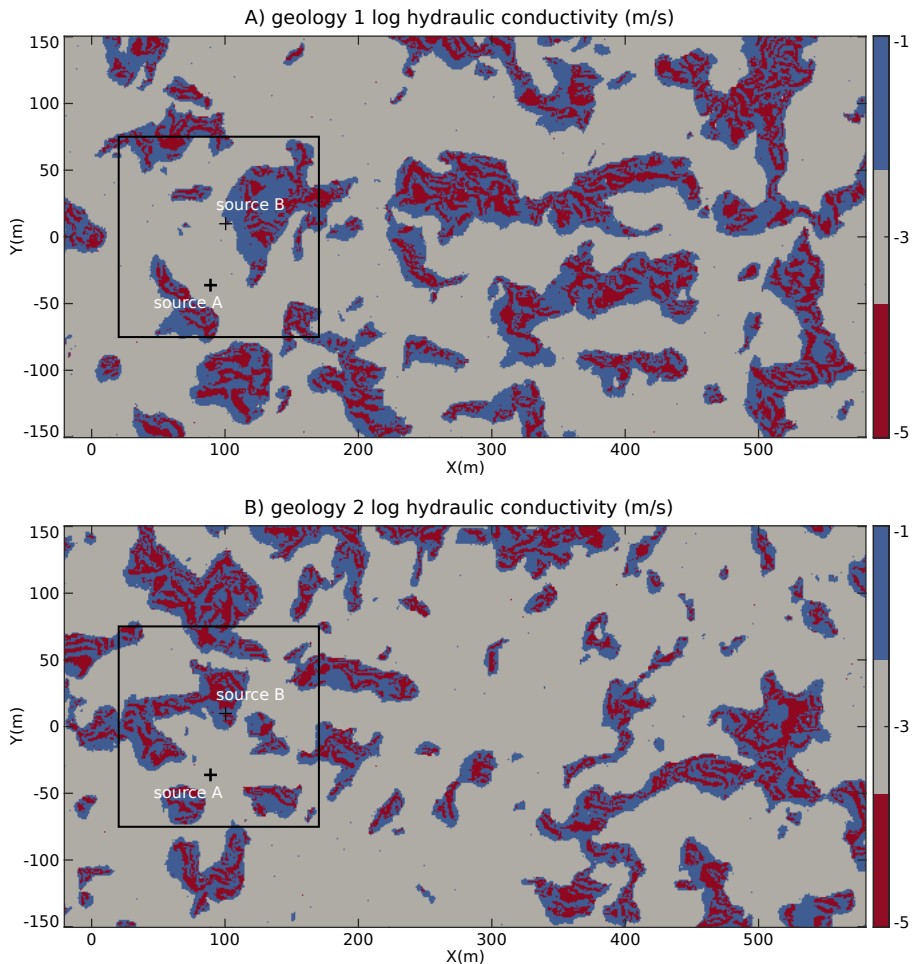

**Figure 1.** Experimental setup: 600m×300m 2D facies model of the aquifer; A) geology 1 and B) geology 2. The black square delimits the possible locations for the search of the contaminant source. The two reference source locations are identified by black crosses.

training image described in Appendix (Figure A1). The hydrogeological properties associated to the facies are given in Table 1 and are inspired from analogs described in the literature (Jussel et al., 1994; Bayer et al., 2011). Note that the contaminant spreading at the scale of this model is assumed to be mainly controlled by the geological heterogeneity. Since there is always

| facies | hydraulic conductivity $K(m/s)$ | porosity | storage coefficient $S_s(m^{-1})$ | molecular diffusion $D_m(m^2/s)$ | longitudinal dispersivity $\alpha_L(m)$ | transversal dispersivity $\alpha_{Th}(m)$ |
|---|---|---|---|---|---|---|
| coarse sediments | $10^{-1}$ | 0.2 | $10^{-5}$ | $10^{-9}$ | 1 | 0.1 |
| mixed sediments | $10^{-3}$ | 0.2 | $10^{-5}$ | $10^{-9}$ | 1 | 0.1 |
| fine sediments | $10^{-5}$ | 0.2 | $10^{-5}$ | $10^{-9}$ | 1 | 0.1 |

**Table 1.** Hydrogeological parameters

a some numerical dispersion when solving the advection dispersion equation numerically, we used the smallest possible value for the longitudinal and transverse dispersivities that would stabilize the numerical problem. Another method to obtain 2D horizontal models of braided river aquifers from 3D models would have been to integrate vertically the hydraulic conductivity field, but since this smoothes out the hydraulic conductivity, the resulting 2D models present less contrasts and less realistic

connected structures.

As boundary conditions for the flow and transport model, we impose a differential head of 2 m on the length of the model (between X= $-20$ m and X= $580$ m) and no flow on the sides (Y= $-150$ m and Y= $150$ m) parallel to the main flow direction. We assume steady-state flow conditions (Figure 2) to run transport simulations by solving the Advection Dispersion Equation with the finite element code Groundwater (Cornaton, 2007).

The source of the contaminant is supposed to be unique, parameterized by the coordinates of its initial center of mass, and located within a search zone delimited by a 150 m $\times$ 150 m square-domain whose coordinates belong to $[20, 170] \times [-75, 75]$. To test the influence of the source location versus the geology, first on the misfit objective function and second on the ability of the proposed approach to deal with more or less complex objective functions, two reference locations ($A$ and $B$) were chosen. Source $A$ is located at ($x_s^A = 89, y_s^A = -36$). Source $B$ is located at ($x_s^B = 100, y_s^B = 10$). Since surface spills usually

present some diffusion characteristics in their shape and can cover different geological features, the initial contaminant mass distribution at time 0 is chosen as a multi-Gaussian distribution centered on the source location with a standard deviation ($\sigma_x = 2.5$ m, $\sigma_y = 1.0$ m) for a total mass $m = 100$ kg. The reference concentration curves $c_{obs}(i,t)$ are obtained for $i = 1, \cdots, 25$ groundwater monitoring wells (Figure 3) and for times $t = 1, \cdots, T$ days. Three concentration breakthrough curves recorded at the well number 2, 16, and 22 are given as examples at the bottom of Figure 3.

Real applications are always characterized by measurement errors. In our practical application of concentration measurements, as for chemical analysis, the errors are mainly due to data acquisition, sampling in the field, dilution procedure, etc. These errors can be assumed either with homogeneous variance or with a standard deviation proportional to the noiseless measurements, e.g. with a proportionality factor supposed to be below $10\%$ (Ramsey and Argyraki, 1997). We denote by $c_{real}(i,t)$ the actual concentration at well $i$ and time $t$ ($1 \leq i \leq 25$ and $1 \leq t \leq T$), i.e. the one that corresponds with the observed con-

centration $c_{obs}(i,t)$ in the noiseless case. Now, for $c_{obs}$, let us assume in the present noisy case that measurements are corrupted with a proportional Gaussian noise, so that observed concentrations become random with

$$c_{obs}(i,t) = c_{real}(i,t) \times (1 + \kappa \, \varepsilon(i,t)), \tag{1}$$

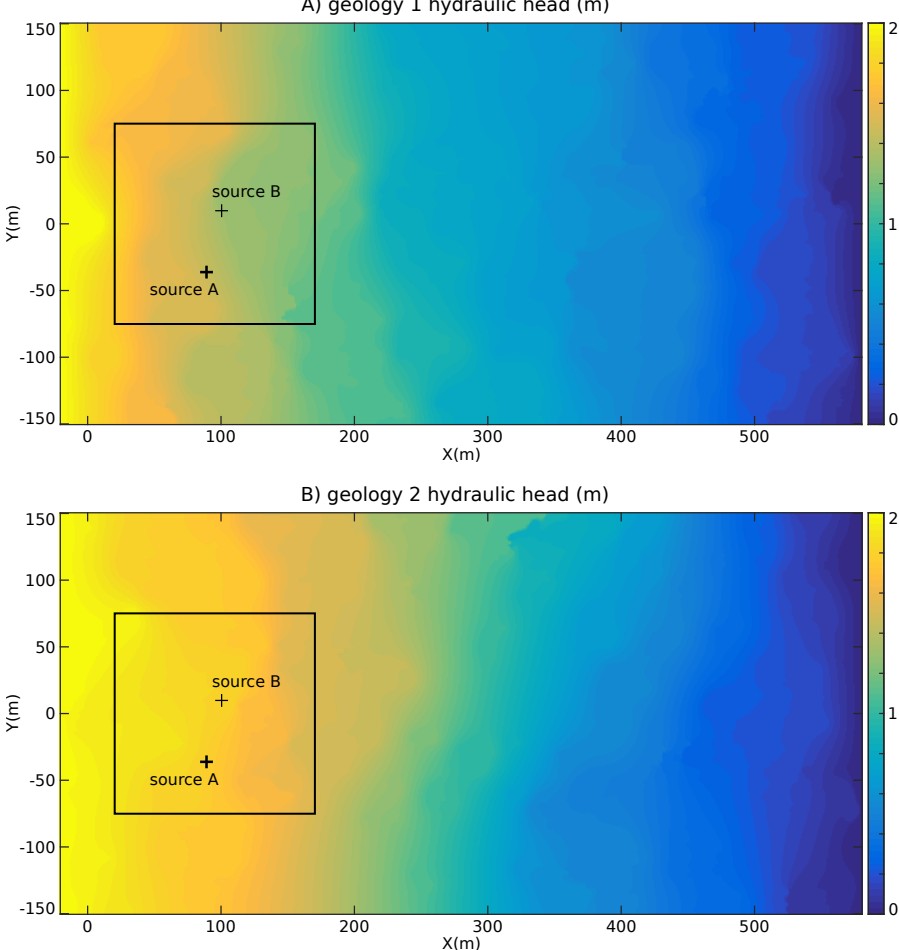

**Figure 2.** Steady state flow for A) geology 1 and B) geology 2. The black square delimits the possible locations for the search of the contaminant source. The two reference source locations are identified by black crosses.

where $\varepsilon(i,t)$ are independent and identically distributed from $\mathcal{N}(0,1)$ and $\kappa$ is a constant such that the level of errors does not exceed a certain proportion.

The unknown location of the contaminant source is denoted as $\mathbf{x} = (x_s, y_s)$. We define $c_{sim}(\mathbf{x}, i, t)$ as the simulated concentration level obtained at $(i, t)$ when the contaminant source is located at $\mathbf{x}$. The aim is to find $\mathbf{x}$ that minimizes the following
5    misfit objective function:

$$f(\mathbf{x}) = \left( \sum_{i=1}^{25} \sum_{t=1}^{T} |c_{obs}(i,t) - c_{sim}(\mathbf{x}, i, t)|^p \right)^{\frac{1}{p}}, \tag{2}$$

which corresponds to the $\ell^p$ distance between the matrices $(c_{obs}(i,t))_{i \in \{1,...,25\}, t \in \{1,...,T\}}$ and $(c_{sim}(\mathbf{x}, i, t))_{i \in \{1,...,25\}, t \in \{1,...,T\}}$, where $p \geq 1$ is a parameter that can be arbitrarily chosen by the modeller (in our experiments both $p = 1$ and $p = 2$ were con-

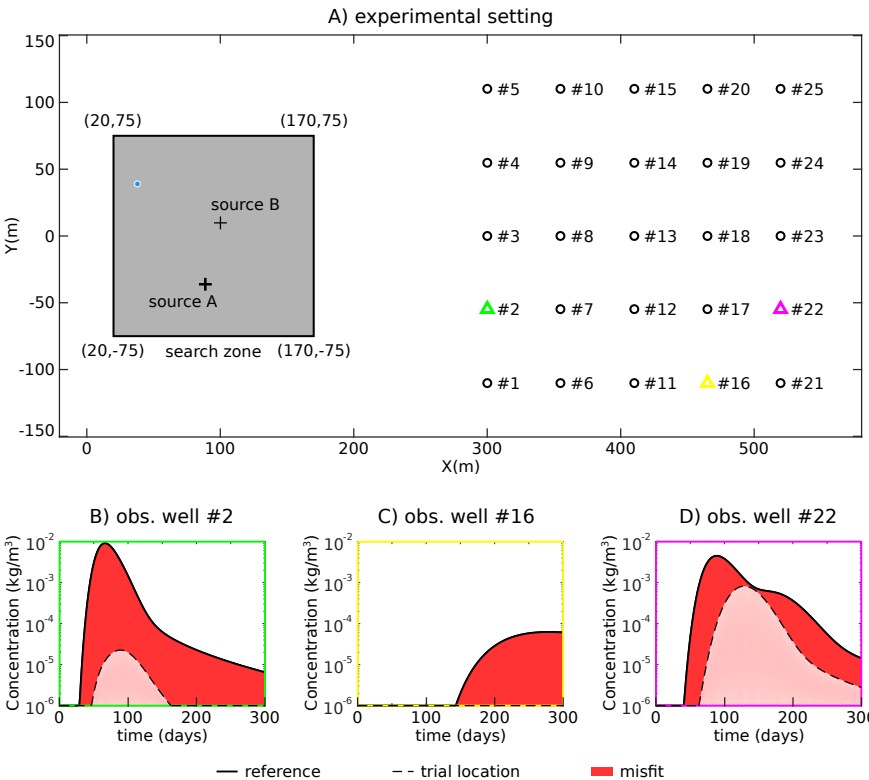

**Figure 3.** Misfit objective function settings; A) Location of the search zone (grey area), of the two reference contaminant sources and of the 25 groundwater monitoring wells (denoted by a circle or a triangle) within the hydrogeological model boundaries; the blue dot denotes the trial location of the contaminant; B), C) and D) misfit components at wells 2, 16, and 22 respectively, resulting from the comparison of the concentration breakthrough curves simulated at the trial location with the recorded ones for reference source A.

sidered, as mentioned later). At the location of the reference source, the function reaches its minimum. In this synthetic study, we neglect conceptual or numerical errors in $c_{sim}$ that may result from an incomplete knowledge of the hydraulic conductivity field or boundary conditions, which would be important to consider in a real field application.

The search zone is restricted to a discrete domain $Z$, using a regular grid of 3 m resolution for three reasons. First, in practical applications, the location of the source is often restricted to an area thanks to historical information about industrial activities or accidents. Here, we apply the same principle but assume a simple geometry. Second, this procedure and geometry allows us to provide an exhaustive computation of the objective function for the research community. Third, it is an interesting problem because most available optimization programs work either on continuous domains or are dedicated to specific classes of optimization problems (Integer programming, mixed linear integer programming), and few seem to be available for non-linear optimization over finite sets beyond metaheuristics used in combinatorial optimization (Rios and Sahinidis, 2013). In the case of our contaminant source localization problem, by the nature of the problem, we have a continuous structure (objective

function) where the domain is restricted to grid points. As an exhaustive evaluation of the objective function over $Z$ is computationally expensive (depending on the mesh resolution), the aim of the optimization is to minimize the objective function $f$ in the search zone within a limited number of iterations and for that purpose, we propose using an EI algorithm.

## 3 Benchmark case generator

An ensemble of time varying concentrations at 25 observation wells is provided at a full factorial design of candidate points in the search zone $Z$, plus at contaminant source location $B$ (source location $A$ belongs to the factorial design), for 2 geological geometries. Allowing any combination of observation wells among the 25, or any source location among the full factorial design, leads to $2^2 \times 2602 \times (2^{25} - 1)$ possible test functions (i.e. more than $349 \times 10^9$ test cases). Moreover, any customized source of error can be added in the generation of the objective function. As these functions are known through their respective
$51^2$ values at the discretized source space $Z$, they can be re-interpolated (e.g. using splines) for continuous optimization purposes. Here we instead consider the discrete problem of selecting the optimal location among $51^2$ candidates and for that goal, we will apply a straightforward discretized version of an EI algorithm as presented in the next section. The data and some R functions to generate benchmarks for any input parameters are provided on GitHub at https://github.com/gpirot/BGICLP. A brief description of the repository is given in Appendix B of this paper.

## 4 Optimization methodology

The optimization algorithm used hereafter to minimize $f(\mathbf{x})$ over the domain relies on a machine learning approach relying on Gaussian Process (GP) models Rasmussen and Williams (2006) to improve iteratively the knowledge of $f(\mathbf{x})$ over the domain. It relies on the iterative evaluation of $f(\mathbf{x})$ at locations whose potential to improve the minimum among the evaluated objective function at previously explored locations is the greatest. The following steps give an overview of the proposed algorithm.
In what follows, more details are given about the required assumptions, the way to estimate $f(\mathbf{x})$ and the definition of the Expected Improvement criterion.

The algorithm belongs to a class of Bayesian optimization algorithms (Mockus, 1989; Shahriari et al., 2016). The Bayesian aspect refers to placing a random process prior $Y$ on the unknown function $f$ (possibly computationally expensive) and updating its probability distribution thanks to available evaluation results. The optimization part relies on using conditional distributions of $Y$ to iteratively choose points with the identification of $f$'s global optimum/optimizer(s) in view. The crux is to fit
adequate probabilistic models and also to design adapted *acquisition functions* (a.k.a *infill sampling criteria* in surrogate-based optimization) in order to drive algorithms to an efficient optimization.

GPs constitute a very popular class of probabilistic models that are fully specified by a mean function $m(\mathbf{x})$ and a covariance function $k(\mathbf{x}, \mathbf{x}')$ Rasmussen and Williams (2006). In this work, we use ordinary kriging with a Matérn ($\nu = 3/2$) covariance
function (See Roustant et al. (2012) for details) and the kernel parameters are estimated by maximum likelihood using the DiceKriging R package. While it is also possible to use a transformation of the response in GP-based optimization (e.g. Jones

**Algorithm 1** Optimization algorithm overview; $n_0$ is the number of initial locations used to define the initial knowledge; $N$ is the budget, or the number of time the objective function can be evaluated; $n$ counts the number of times that the objective function has been evaluated.

---

Knowledge initialization: evaluate $f(\mathbf{x})$ at $n_0$ initial locations defined by an initial design

Set $n = n_0$

**while** $n \leq N$ **do**

    Based on the current knowledge, compute the Expected Improvement criterion $\text{EI}_n(\mathbf{x})$ over the domain

    Evaluate $f(\mathbf{x})$ where $\text{EI}_n(\mathbf{x})$ is maximum

    Increment the knowledge and $n$

**end while**

Return the location where $f(\mathbf{x})$ is minimum over the evaluated locations

---

et al., 1998a), on the considered data it did not lead to substantial differences in optimization performance despite the non-negativity of the misfit.

Denoting training inputs and outputs as $\mathbf{X}_n = (\mathbf{x}_1, \mathbf{x}_2, \ldots, \mathbf{x}_n)$ and $\mathbf{f}_n = (f(\mathbf{x}_1), f(\mathbf{x}_2), \ldots, f(\mathbf{x}_n))$, assuming a GP prior with a constant unkown mean (endowed with an improper uniform prior) leads to a Gaussian conditional distribution with the following marginal predictive mean and variance:

$$m_n(\mathbf{x}) = \hat{\mu} + \mathbf{k}(\mathbf{x})^T \mathbf{K}^{-1}(\mathbf{f}_n - \hat{\mu}\mathbf{1}) \tag{3}$$

$$s_n^2(\mathbf{x}) = k(\mathbf{x}, \mathbf{x}) - \mathbf{k}(\mathbf{x})^T \mathbf{K}^{-1}\mathbf{k}(\mathbf{x}) + \frac{(1 - \mathbf{k}(\mathbf{x})^T \mathbf{K}^{-1}\mathbf{1})^2}{\mathbf{1}^T \mathbf{K}^{-1}\mathbf{1}}, \tag{4}$$

where $\mathbf{K} = (k(\mathbf{x}_i, \mathbf{x}_j))_{i,j=1,\ldots,n}$ is the $n \times n$ prior covariance matrix (assumed invertible here) of responses at training inputs, $\mathbf{k}(\mathbf{x}) = (k(\mathbf{x}, \mathbf{x}_1), \ldots, k(\mathbf{x}, \mathbf{x}_n))^T$ is an $n \times 1$ covariance vector and $\hat{\mu} = \frac{\mathbf{1}^T K^{-1}\mathbf{f}_n}{\mathbf{1}^T \mathbf{K}^{-1}\mathbf{1}}$ is the best linear unbiased estimate of $\mu$.

The optimization algorithm typically starts with constructing a space-filling design $\mathbf{X}_{n_0} = (\mathbf{x}_1, \mathbf{x}_2, \ldots, \mathbf{x}_{n_0})$ (See, e.g., (Dupuy et al., 2015)) and evaluating $f(\mathbf{X}_{n_0})$ to initialize the knowledge of the algorithm (e.g., $n_0 = 9$ blue dots in the left panel of Figure 4A). Here the initial $\mathbf{X}_{n_0}$ is generated based on latin hypercube sampling (McKay et al., 1979). Then, the algorithm begins its iterations. In each iteration, the ensemble of $n$ available evaluations $\mathbf{f}_n = (f(\mathbf{x}_1), f(\mathbf{x}_2), \ldots, f(\mathbf{x}_n))$ is used to train the GP model and make predictions at yet unexplored decision space locations. The predictive distribution is then used to compute the so called Expected Improvement criterion (Mockus, 1989), which indicates at every point in the decision space how much the objective function value may be decreased relative to $f_{\min} = \min \mathbf{f}_n$, in expectation:

$$\text{EI}_n(\mathbf{x}) = \mathbb{E}_n \left[ \max\left(0, f_{\min} - Y(\mathbf{x})\right) \right]. \tag{5}$$

The EI criterion offers a good balance between exploitation of regions with low predictive mean values and exploration of regions with high predictive means, which provides an efficient optimization search scheme (e.g., red dot in the right panel of

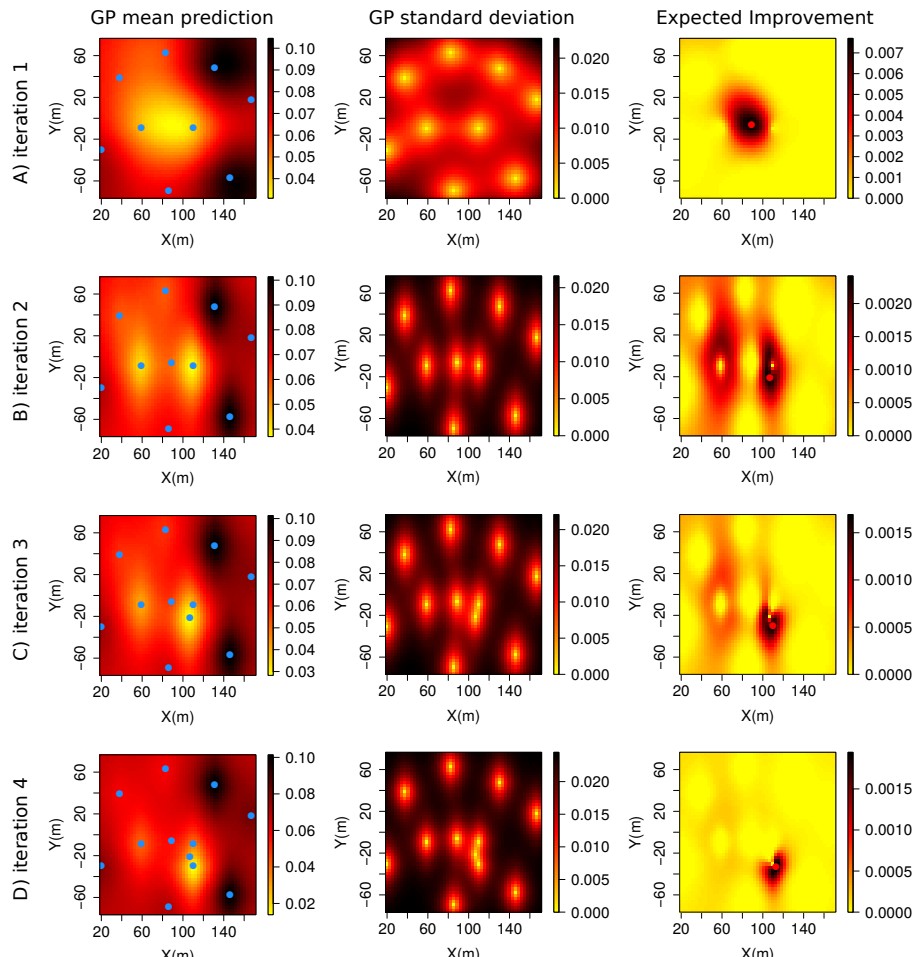

**Figure 4.** Illustration of the first four EI algorithm iterations for scenario 1; the sub-figures in the left column illustrate the prediction mean of $f$ over the two-dimensional decision space at each iteration; the blue dots indicate the decision space locations where $f$ was previously evaluated; the sub-figures in the center column illustrate the prediction variance of $f$ over the two-dimensional decision space at each iteration; the sub-figures in the right column illustrate the expected improvement map over the two-dimensional decision space at each iteration; the red dot denotes the decision space location with the maximum EI value.

Figure 4A). It turns out that EI can be calculated analytically (Mockus, 1989; Jones et al., 1998b). In our discrete settings with moderate number of search points, the EI can be computed at all unevaluated locations of $f$ (e.g. right panels of Figure 4). The decision space location with the largest EI value is considered as the next point $\mathbf{x}_{n+1}$ (e.g. red dot on right panels of Figure 4) to evaluate $f$. The optimization is run using the DiceKriging and DiceOptim R packages developed by Roustant et al. (2012). The number of iterations is fixed in advance (91 in what follows) so that it stops when the maximum number of iterations allowed is reached. Covariance parameters are updated after each iteration by Maximum Likelihood Estimation.

## 5   Results

The results for both the noiseless and noisy cases are presented in this section. The main results are presented in Section 5.1. They rely on using information from all wells, and on noiseless concentration observations for the 4 configurations engendered by 2 geological scenarios and 2 possible sources of contaminant. For completeness, the algorithm sensitivity analysis with the noise added to the objective function and with various well configurations are presented in Section 5.2.

Note that with an initial space-filling design of $n_0 = 9$ elements, and a number of iterations of 91, we define here a total budget of $N = 100$ evaluations of the objective function.

### 5.1   Main results for noiseless cases

Using information from the 25 observation wells, the optimization algorithm is applied over 4 configurations that depend on the retained geology and on the contaminant source location as described in Table 2, where the noise level $\kappa$ (of Eq. 1) is set to 0 and the parameter $p$ of the objective function $f(x)$ is set to 2. Starting from a specific initial design, the explorations

| case | type of geology | source coordinate |
|------|-----------------|-------------------|
| 1 | geology 1 | $(89, -36)$ |
| 2 | geology 1 | $(100, 10)$ |
| 3 | geology 2 | $(89, -36)$ |
| 4 | geology 2 | $(100, 10)$ |

**Table 2.** Description of the 4 configurations.

of the objective functions by the EI algorithm (aiming at the contaminant source localization), are displayed in Figure 5 for each scenario. These objective functions display multiple local minima, narrow valleys and sometimes very flat bottoms. These characteristics make the search for the global minimum challenging especially for gradient based techniques. The locations explored by the EI algorithm are plotted over the 3 m × 3 m discretization of the objective function $f$. The white and blue dots represent respectively the initial and then explored locations where the objective function is evaluated by the algorithm. In most cases, the minimum of the discretized objective function is reached in less than 50 evaluations. The geology seems to be the dominating factor for the global patterns of the objective function. Note that for scenarios 2 and 4, the contaminant source is located at $(100, 10)$, which is not within the discretized grid of the objective function; the closest point on the discretized grid is $(101, 9)$. For scenario 2, the minimum of the objective function is less than 3 m apart from the reference source located at $(100, 10)$. However, for scenario 4, the reference source located at $(100, 10)$ and the minimum of the objective function located at $(80, 18)$ are 25 m apart.

The performance of the optimization algorithm is assessed on 100 runs of the algorithms. Each run is characterized by a specific and uniformly drawn 9-point initial design. Each run is allowed a total budget of 100 evaluations of the objective function. The performance depends on the number of iterations required to locate the minimum of the objective function $\min_{\mathbf{x}} f(\mathbf{x})$. The performance can be assessed directly by looking at the optimality gap, i.e., the distance between the location of the best estimated minimum $f_{\min}$ of the objective function and the location of its minimum $\min_{\mathbf{x}} f(\mathbf{x})$ as a function of

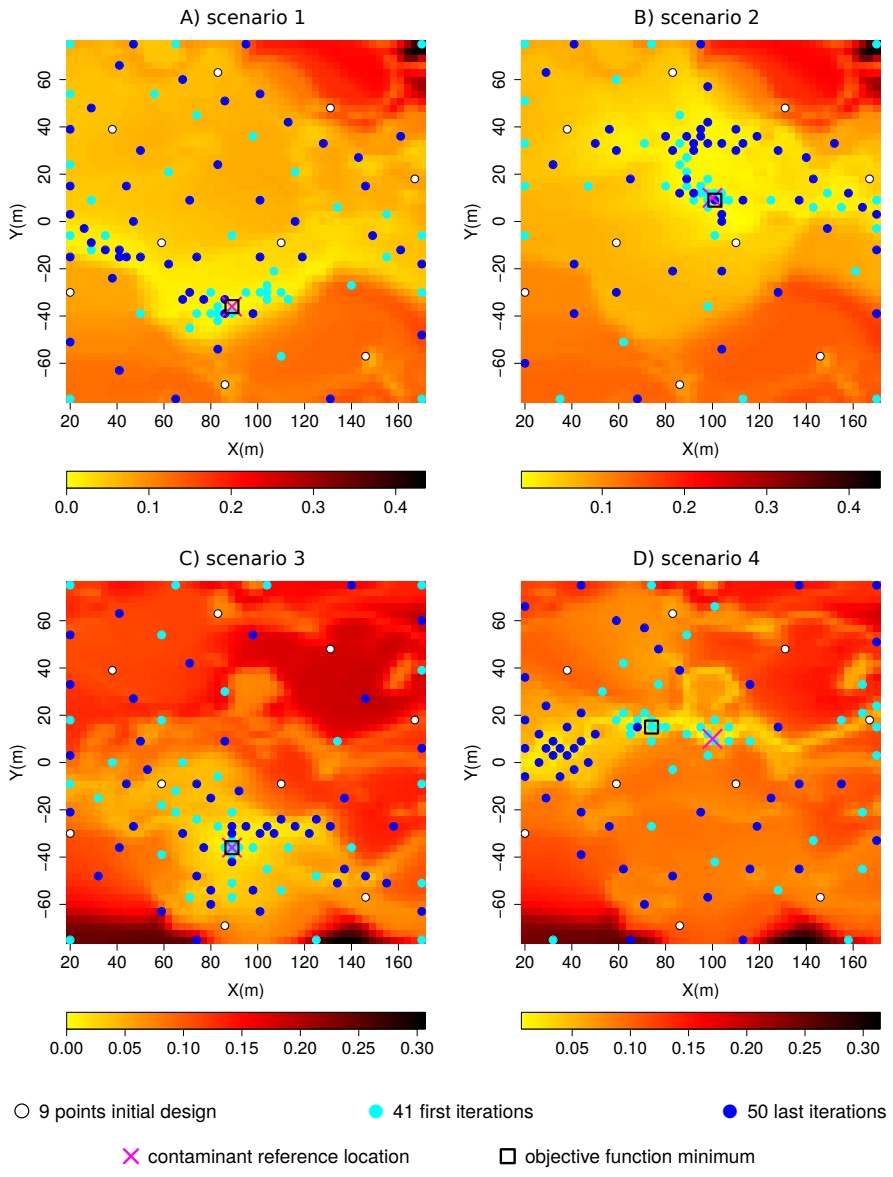

**Figure 5.** Solution exploration results for the 4 scenarios over the cost functions; A & B for geology 1; C & D for geology 2; A & C for initial contaminant location at $(89, -36)$; B & D for contaminant initial location at $(100, 10)$.

the number of evaluations of $f$ (Figure 6A-D). Another possibility is to look at the normalized best found minimum misfit between the true minimum $\min_{\mathbf{x}} f(\mathbf{x})$ and the best estimated minimum of the objective function $f_{\min}$ as a function of number of evaluations of $f$ (Figure 6E-H). Both indicators behave similarly. Finally, the performance of the localization algorithms can be assessed by analyzing the distribution of the distance of the explored location that is closest to the true contaminant source

over the 100 runs for a given number of iterations (Figure 7). Independently from the considered scenario, the bin counts for lowest values significantly increase when the number of iterations increase, and the bin counts for distances over 20 m rapidly come down to 0.

## 5.2 Sensitivity of the algorithm performances to errors and to well configuration

In what follows, we show the results of a joint sensitivity analysis of the algorithm performance to proportional measurement

errors and to the number of well retained in the computation of the objective function. Four levels of proportional measurement errors are tested: $0\%$, $10\%$, $20\%$ and $40\%$. Seven well configurations with 1, 3, 5, 10, 15, 20 or 25 wells are tested. The identification of the wells for each configuration is given in Table 3. The cross-joint sensitivity analysis is then composed of 28

| number of wells | well id |
|---|---|
| 1 | 13 |
| 3 | 11,13,15 |
| 5 | 11,12,13,14,15 |
| 10 | 11,12,13,14,15,1,2,3,4,5 |
| 15 | 11,12,13,14,15,1,2,3,4,5,21,22,23,24,25 |
| 20 | 11,12,13,14,15,1,2,3,4,5,21,22,23,24,25,6,7,8,9,10 |
| 25 | 1 to 25 |

**Table 3.** Description of the 7 well configurations.

scenarios. The resulting objective functions are illustrated in Figure D1. One can note that, the precision becomes finer around the true minimum of the objective function, when increasing the number of wells. However, the improvement is limited once

a line of 5 wells, orthogonal to the main flow direction, is used. The concentration measurement errors, even if proportional to $40\%$, have a negligible impact on the objective function. For each scenario, the algorithm is run 100 times. Each run is characterized by a specific and uniformly drawn 9-point initial design. Each run is allowed a total budget of 100 evaluations of the objective function. The optimality gaps, showing the performance of the algorithm for the 28 scenarios, are displayed in Figure D2. The optimality gap is improved with an increasing number of wells (until a full column of wells is used) and not

affected by concentration measurement errors.

## 6 Discussion

Through successive kriging of the misfit between simulated and observed concentrations, guided by the expected improvement criterion, the proposed optimization algorithm localizes efficiently the source of a contaminant in a 2D geological environment representing realistic patterns and property contrasts. The algorithm requires approximatively 50 evaluations of the objective

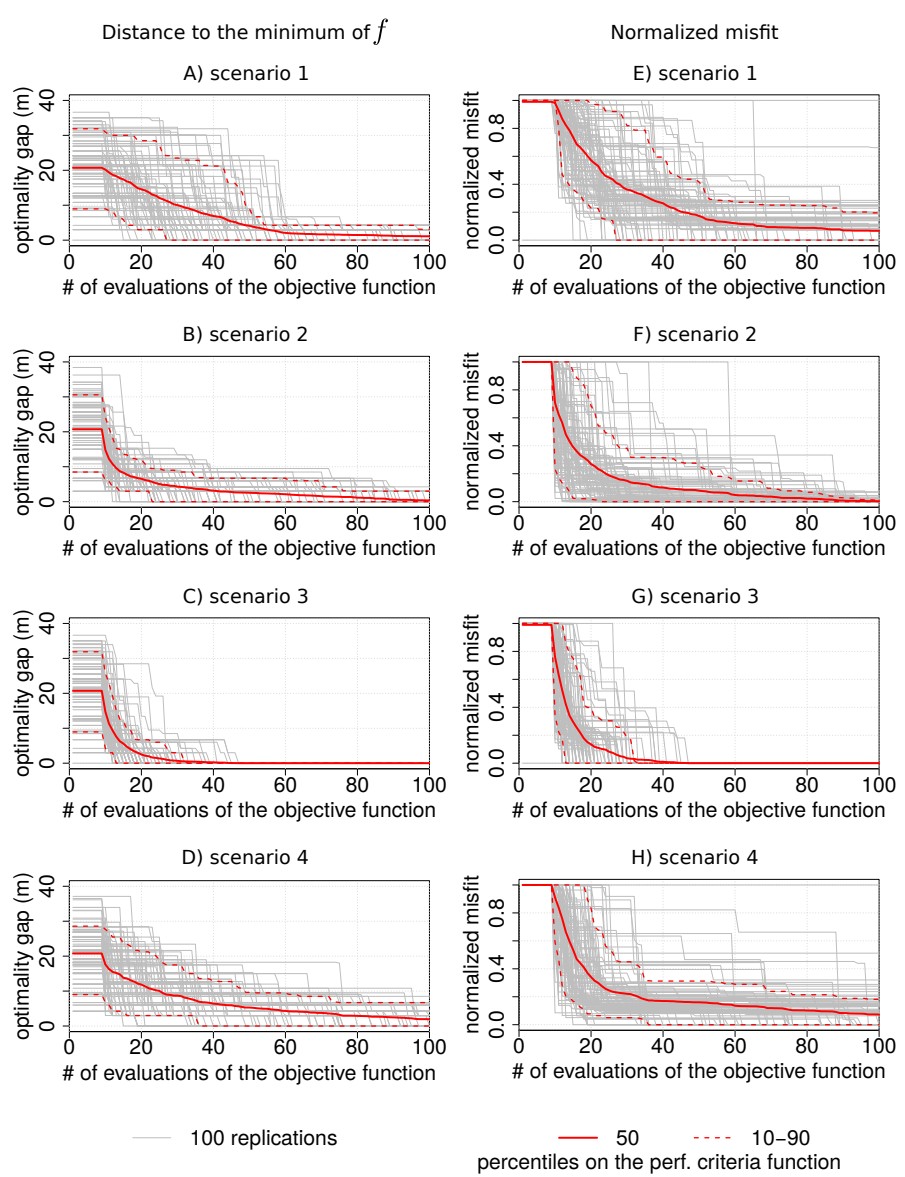

**Figure 6.** Performances of the EI optimization algorithm as a function of number of evaluations of the objective function for 100 different initial design; A), B), C) & D) distance of the best solution to the location of the objective function minimum; E), F), G) & H) normalized misfit; A) & E) scenario 1; B) & F) scenario 2 ; C) & G) scenario 3 ; D) & H) scenario 4.

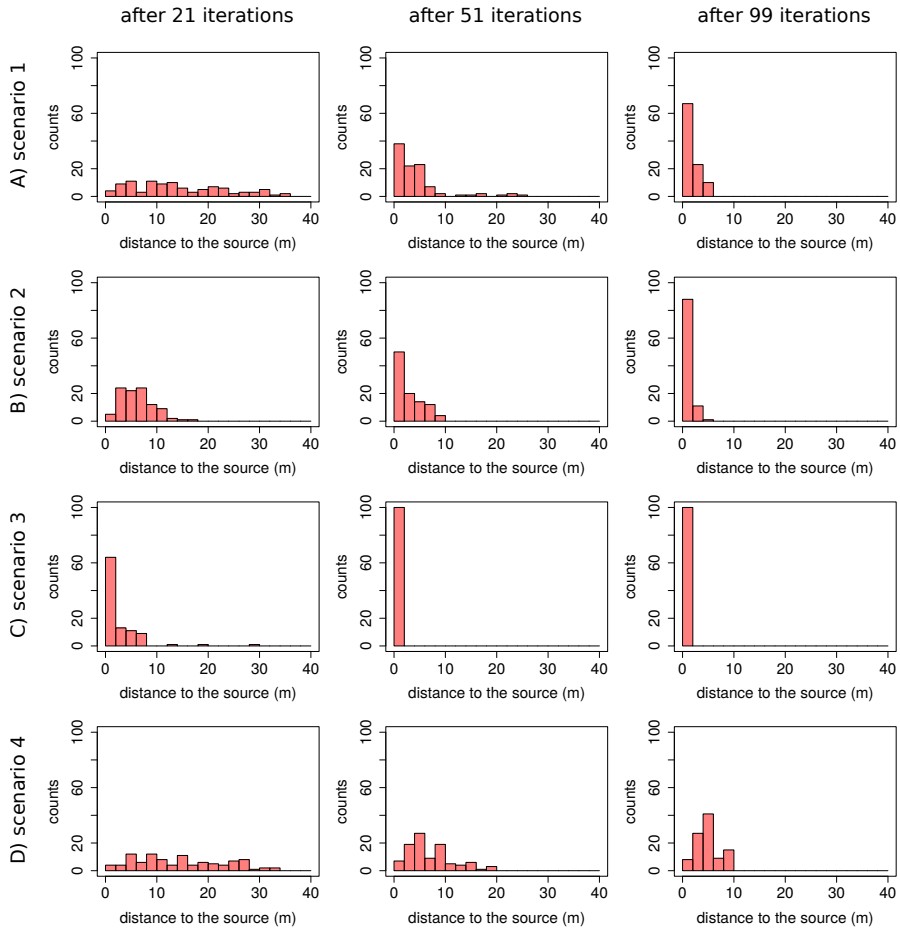

**Figure 7.** Distance to the contaminant source distribution for 100 runs for the best solution given by the EI algorithm ; row A) to D) for scenarios 1 to 4.

function in comparison to more than 2600 for an exhaustive evaluation of the discretized search zone ($\sim 1.9\%$). The total number of candidate points would increase exponentially in the number of dimensions of the parameter space, eliminating exhaustive search as an option, from even moderate dimensions, when assuming a high resolution.

Comparison of the different scenarios reveals that the geology controls the main features of the objective functions, which reinforce the importance of realistic geological structures in contaminant source localization problems. Of course, the shape and location of lower value zones of the objective functions are controlled by the reference location of the contaminant source. The results presented here are based on an objective function $f$ computed with $p = 2$, which corresponds to an $\ell^2$ distance between reference and candidate concentration values (See Eq. 2). As the choice of $p$ may substantially influence the flat or deep aspect of valleys (low value zones) of the objective function, we additionally tested the EI algorithm on the 4 scenarios for objective functions with $p = 1$. We found that building $f$ onto the $\ell^2$ distance leads to flatter wide valleys of low values for

the objective functions, which might not favor the efficiency of the EI optimizer. However, the results and performances of the EI algorithm are very similar between the two norms tested. This is why we decided not to show the results of the algorithm objective functions built upon the $\ell^1$ distance.

When proportional measurement errors do not exceed 10, 20, 30 or $40\%$, the objective function is quasi identical and the algorithm performance is not affected. It is not surprising as the objective function is a mean of the misfit over several monitoring locations and time, which contributes to filter out the error, except for a positive bias. However, for other applications, the resulting noise in the objective function might require a more specific treatment, e.g. appealing to strategies adapted to deal with noisy function evaluations (See for instance Picheny et al. (2013); Picheny and Ginsbourger (2014a) for an overview and tutorials based on R code). Here we consider measurement errors that are proportional to the actual concentrations. However, it might take a different form. In Appendix C, we propose a more general definition of possible Gaussian measurement errors and derive the resulting objective function covariance matrix.

An interesting result is that the number and configuration of wells has a strong impact on the objective function until a "full" line of wells, orthogonal to the main flow direction, is used. Increasing the number of wells on an axis orthogonal to the main flow direction improves greatly the characterization of the objective function, notably around the true contaminant source. It confirms what is often done in practice to catch contaminant plumes. Adding another line of observation wells seems less promising than densifying a column of wells. Of course, densification might be limited in practice by minimum distances between wells to avoid connecting artificially separated flowpaths for instance, but depending on the level of site characterization, a similar algorithm could then be used to optimize well configurations.

By making the source code of the objective function generator available for public use, we provide several benchmark objective functions. These latter are driven by real hydrogeological applications and can be used for testing and comparing optimization techniques. This benchmark will fill a gap for the community of applied mathematicians and statisticians who develop optimization algorithms and who want to test their tools on realistic objective functions. In addition, hydrogeologists will benefit from the code provided in the GitHub repository so that they can implement the proposed optimization algorithm in their own applications. For the test case documented here and given the structure of the objective functions that are defined on a discrete domain, it does not seem relevant to apply off-the-shelf combinatorial algorithms. However it would be certainly of interest to compare the proposed approach to genetic/evolutionary algorithms compatible with such settings. A pragmatic approach here, to enable comparisons with a broader class of derivative-free and also derivative-based algorithms, would be to re-interpolate the data (with a careful inspection of the optima of the interpolator, i.e. a check that it is not perturbing the problem by too many potential artifacts) and conduct a benchmark involving Bayesian optimization (with EI and potentially also other infill sampling criteria) against a selection of state-of-the-art algorithms.

Strong assumptions have been made to localize the contaminant source in the presented application. The hydrogeolological properties and the flow boundary conditions are assumed to be perfectly known and the hydrogeological model is spatially limited to two dimensions. This allowed to compare the outcome and efficiency of the algorithm with respect to a full grid search of the objective function. Because of their expensive computing costs to assess the objective function at one location of the parameter space, three-dimensional applications will not allow for an exhaustive search of the solution; this is why

they may require, in the near future, optimization algorithms such as the one proposed in this paper. Further research should also consider the uncertainty related to hydrogeological property characterization and flow and transport boundary conditions. Some steps have already been made in that direction (Koch and Nowak, 2016), but were limited to multi-Gaussian conductivity fields. In addition, a regular grid discretization might compromise the ability to accurately locate the contaminant source in the presence of a strong flow path. For example, in a real-world application, the contaminant source has a very low probability of being located on a grid node. This problem could be avoided by using adaptive meshing, which would require more computing resources.

## 7   Conclusions

The use of 2D hydraulic conductivity fields that present sharp contrasts and specific connectivity patterns produces complex objective functions with multiple local minima. The proposed benchmark tool produced from these complex functions offers challenging real-world test for developers of optimization algorithms. The EI algorithm used in this 2D study localized efficiently the contaminant source that is located on a grid node. More generally, the proposed algorithm is an interesting approach for combinatorial optimization algorithm. The objective functions and the performance of the algorithms are not affected by proportional measurement errors lower than $10\%$ (even $40\%$). The objective function is strongly determined by the geology and by the monitoring well configuration (number and location). In particular, the characterization of the objective function, on which the performance of the algorithm rely, is greatly improved when a line of monitoring wells orthogonal to the main flow direction is densified. To improve the limitation imposed by a source centered on the nodes of a fixed mesh, which is independent of the optimization algorithm, future research could be conducted on optimization embedding adaptive meshing in flow and transport simulations; another possibility would be to relax the constraint on mass distribution of the initial plume as a way to deal with its related uncertainty. The effective performance of the algorithm on this 2D case is encouraging to continue toward 3D applications and toward integration of geological uncertainty in contaminant source localization problems.

*Code and data availability.*  The data and some R functions to generate benchmarks for any input parameters are provided on GitHub at https://github.com/gpirot/BGICLP. A brief description of the repository is given in the Appendix of this paper.

## Appendix A:  Training Image

## Appendix B:  Supplementary material

The electronic supplementary material provided on the GitHub repository at https://github.com/gpirot/BGICLP with this paper contains 3 folders and 2 R-scripts.

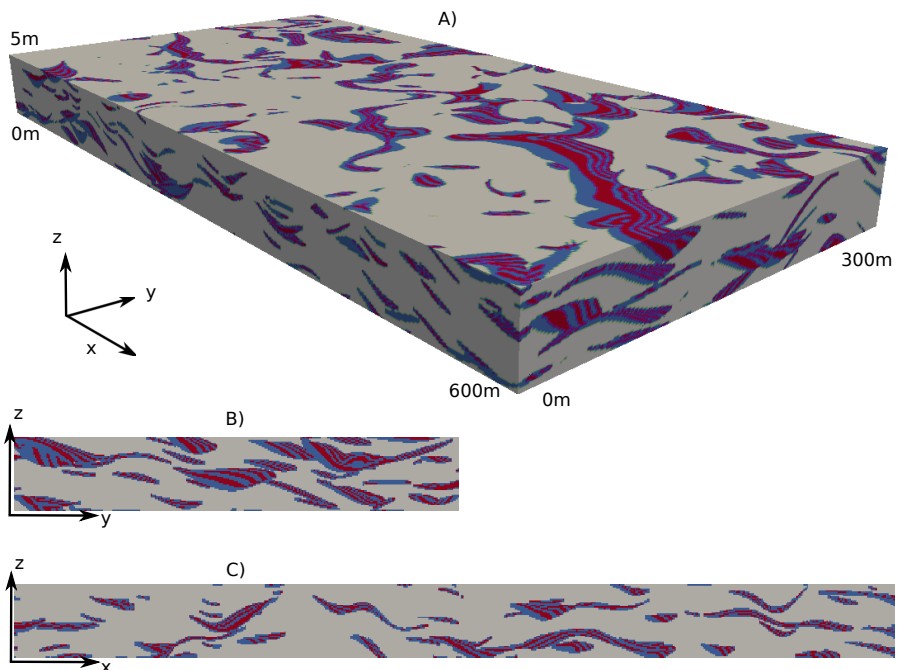

**Figure A1.** 600m ×300m ×5m training image with vertical scale exaggerated by 10; A) 3D representation; B) vertical section transversal to the main flow direction; C) vertical section longitudinal to the main flow direction. This three-dimensional model was generated by a pseudo-genetic algorithm proposed by Pirot et al. (2015). It is obtained by imitation of successive erosion and deposition events. Successive conditional simulations of topographies (Pirot et al., 2014) stacked together produce successive layers that are filled by heterogeneous geological facies according to a rule mimicking flow and sedimentation processes.

The 'data' folder contains 1) the simulated concentration $c_{sim}(i,t)$ and the actual concentrations $c_{real}(i,t)$ over $Z$ and the contaminant source locations $A$ and $B$ at $i = 1, \cdots, 25$ observation wells for the 2 geologies , 2) the **x** coordinates of the search zone $Z$ and of the contaminant source locations $A$ and $B$.

The 'figures' folder contains illustrations of $f(\mathbf{x})$ over $Z$ for each of the 4 configurations when considering the 25 wells with

5   the $\ell^2$ norm.

The 'src' folder contains 4 R scripts. The 'image.scale.R' script, created by Pretty R at inside-R.org is used for graphic illustration purposes. The 'generate_lhs_on_grid.R' script allows generating initial point designs by latin hypercube sampling. The 'functionAddNoise.R' script defines the measurement error to apply. The 'functionGenerator.R' script takes as arguments a selection of observation wells $\mathcal{W}$, a type of geology, the source coordinates and the type of norm used. It produces the

10   evaluation of the objective function $f(\mathbf{x})$, as defined in Eq. 2.

The 'plotGeneratedFunction.R' script illustrates the use of the function generator and saves the plot in the 'figures' folder. The 'runEGO.R' script gives an example of how to use the proposed optimization algorithm.

## Appendix C: General form of error integration in the objective function

More generally, for $c_{obs}$, one might assume that measurements are corrupted with a Gaussian noise with variance $\sigma(i,t)$ that may depend on both the well $i$ and the time $t$, so that observed concentrations become random with

$$c_{obs}(i,t) = c_{real}(i,t) + \sigma(i,t)\varepsilon(i,t), \tag{C1}$$

where $\varepsilon(i,t) \sim \mathcal{N}(0,1)$. Here for the sake of brevity we assume that the $\varepsilon(i,t)$ are independent for different $(i,t)$ pairs, but the following can be extended without major difficulty to the case of correlated normals with prescribed correlation matrix. Note that from the additive formulation above, a multiplicative noise setting can be obtained by taking $\sigma(i,t)$ proportional to $c_{real}(i,t)$. Imposing for instance $\sigma(i,t) = c_{real}(i,t)$, one gets indeed $c_{obs}(i,t) = c_{real}(i,t)(1+\varepsilon(i,t))$. Let us now focus on the effect of noise on the objective function, and consider for simplicity the squared misfit in the case $p = 2$, which becomes a random function denoted henceforth by $f_{\varepsilon}^2$ while $f^2$ stands for the deterministic squared misfit from the noiseless case. We then have

$$
\begin{aligned}
f_{\varepsilon}^2(\mathbf{x}) &= \sum_{i=1}^{25}\sum_{t=1}^{T} \left(c_{obs}(i,t) - c_{sim}(\mathbf{x},i,t)\right)^2 \\
&= \sum_{i=1}^{25}\sum_{t=1}^{T} \left(c_{real}(i,t) + \sigma(i,t)\varepsilon(i,t) - c_{sim}(\mathbf{x},i,t)\right)^2 \\
&= f^2(\mathbf{x}) + \sum_{i=1}^{25}\sum_{t=1}^{T} \sigma(i,t)^2\varepsilon(i,t)^2 + 2\sum_{i=1}^{25}\sum_{t=1}^{T} \sigma(i,t)\left(c_{real}(i,t) - c_{sim}(\mathbf{x},i,t)\right)\varepsilon(i,t).
\end{aligned} \tag{C2}
$$

A first important note following the expansion above is that the second term, i.e. $\sum_{i=1}^{25}\sum_{t=1}^{T} \sigma(i,t)^2\varepsilon(i,t)^2$, does not depend on $\mathbf{x}$ so that ignoring it would not affect the behavior of optimization algorithms unless they are sensitive to a global shift ( e.g. because of tuning parameters or stopping rules that would depend on the actual values and not solely on relative ones). In our case such a shift is not detrimental, and can even mitigate the potential issue of predicting negative misfits when using GP models without response transformation. For information, up to rescaling, the statistical distribution of this shift belongs to the generalized chi-square family (and to the usual chi-square family in the case of homogeneous $\sigma$). On the other hand, the last term of Eq. C2 does depend both on $\mathbf{x}$ and on the noise $\varepsilon$. Denoting $\eta_{\mathbf{x}} = 2\sum_{i=1}^{25}\sum_{t=1}^{T}\sigma(i,t)\left(c_{real}(i,t) - c_{sim}(\mathbf{x},i,t)\right)\varepsilon(i,t)$, it is then easy to show that $\eta$ defines a centered Gaussian random field indexed by $\mathbf{x}$ in the search domain $Z$, and that the covariance kernel of $\eta$ boils down to the following:

$$\mathrm{Cov}(\eta_{\mathbf{x}}, \eta_{\mathbf{x}'}) = 4\sum_{i=1}^{25}\sum_{t=1}^{T}\sigma(i,t)^2\left(c_{real}(i,t) - c_{sim}(\mathbf{x},i,t)\right)\left(c_{real}(i,t) - c_{sim}(\mathbf{x}',i,t)\right). \tag{C3}$$

In other words, in cases like here when $c_{real}$ is actually known and experiments are ran for benchmarking purpose, it is possible to propagate the effect of noise corruption on the objective function without needing to appeal to the whole set of $c_{sim}$ values at all times and wells, but rather to a pre-calculable covariance matrix from which the error affecting $f$ over the grid search can be simulated. Denoting by $A_{\mathbf{x}}$ the $25 \times T$ matrix of generic entry $\left(2\sigma(i,t)\left(c_{real}(i,t) - c_{sim}(\mathbf{x},i,t)\right)\right)$ and by $\mathbb{1}_j$ a vector

of ones in dimension $j \geq 1$, the covariance kernel of $\eta$ can be written in compact form as $\text{Cov}(\eta_{\mathbf{x}}, \eta_{\mathbf{x}'}) = \mathbb{1}'_{25}(A_{\mathbf{x}} \circ A_{\mathbf{x}'})\mathbb{1}_T$, where $\circ$ stands for the Hadamard (element-wise) product between matrices of identical dimensions.

## Appendix D: Sensitivity to concentration measurement errors and to the number of monitoring wells

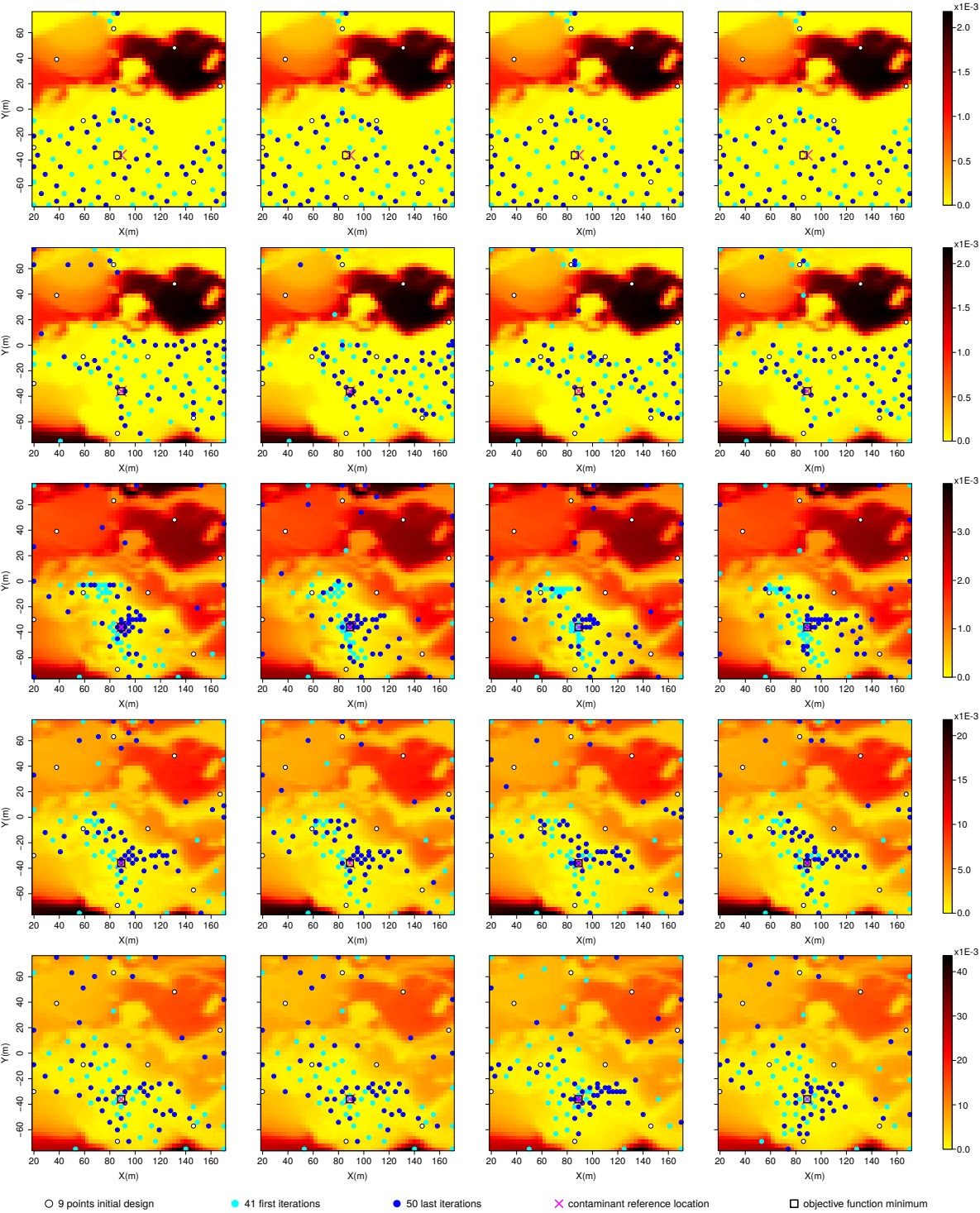

**Figure D1.** Objective function sensitivity analysis; column 1: no noise, column 2: 10% noise, column 3: 20% noise, column 4: 40% noise; row 1: 1 well, row 2: 3 wells, row 3: 5 wells, row 4: 15 wells, row 5: 25 wells.

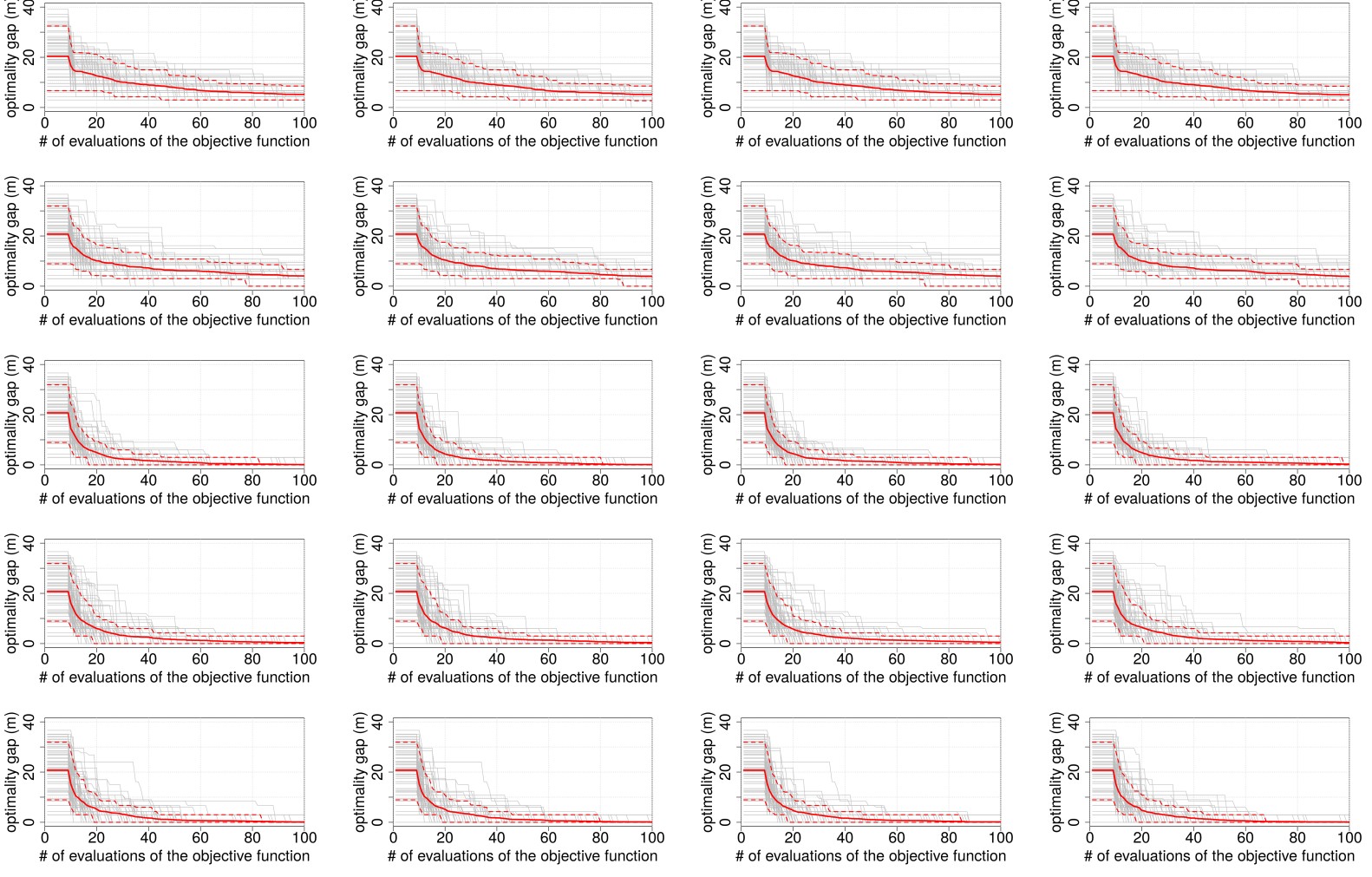

**Figure D2.** Optimality gap sensitivity analysis; column 1: no noise, column 2: 10% noise, column 3: 20% noise, column 4: 40% noise; row 1: 1 well, row 2: 3 wells, row 3: 5 wells, row 4: 15 wells, row 5: 25 wells.

*Competing interests.* The authors declare that they have no conflict of interest.

*Acknowledgements.* The authors would like to thank Fabien Cornaton for his support in the parameterization and use of Groundwater, Emily Voytek and Andrew Greenwood for their support in improving the reading of the manuscript, the anonymous reviewers and the editor Bill Hu for their comments and support. The second author would like to acknowledge support from the Oeschger Center for Climate Change Research (University of Bern), the Swiss Government Excellence Scholarship, as well as the Thailand Research Fund (MRG6080208).

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
