# Peer review of "Contaminant source localization via Bayesian global optimization"

_Hydrology and Earth System Sciences, 2017_

## Referee Comment (RC1) · Anonymous Referee #1 · 12 Sep 2017

The authors present a Bayesian global optimization approach to identify a contaminant source using concentration observations. Two synthetic aquifers with two different log K reference fields are used to demonstrate the efficiency of the approach for the contaminant source localization. Here are certain parts of the manuscript still need improvement for better understanding. Before I can recommend the paper for publication the following comments must be addressed

General comment:

1, In the introduction section, you have introduced some classification for groundwater pollution source identification. In terms of the Bayesian approach you used, you also need to introduce the state of art of contaminant source localization based on Bayesian approaches (eg., Cupola et al. [2015], Zeng et al. [2012] , Raziyeh Farmani et al.

[Figure]

[2009]), and detail the difference and merit of the approach you used by comparing with those Bayesian-based approaches.

2, Why do you choose source A (89, -36) and B (100, 10) as reference contaminant sources? Besides, it is better to show source A and B in Figure 1.

3,On page 10, can you explain why there is large departure between reference source location and minimum of the objective function location.

Specific comments:

1, On page 1, it is better to briefly explain how analytical solution and regression approaches works, just as you have done to the other three categories.

2, Please be careful to use "To the best of our knowledge" on page 2 and 3.

3, It is not appropriate to set longitudinal dispersivity as 1 m when your resolution of the aquifer is 1m*1m.

4, Please show both source A and B in Figure 3 and Figure 1.

5, The dots in Figure 5 are with three different colors, and you just explained the meaning of blue and white dots. Please explain the left one.

---

## Author Comment (AC1) · 20 Sep 2017

The original comments of referee 1 are in black and our answers to these comments are in blue.

**General comments**

1, In the introduction section, you have introduced some classification for groundwater pollution source identification. In terms of the Bayesian approach you used, you also need to introduce the state of art of contaminant source localization based on Bayesian approaches (eg., Cupola et al. [2015], Zeng et al. [2012] , Raziyeh Farmani et al.[2009]), and detail the difference and merit of the approach you used by comparing with those Bayesian-based approaches.

The term 'Bayesian' is typically used when a method involves Baye's rule. And, as the referee points out, many Bayesian approaches have been used throughout groundwater sciences and notably for contaminant source localization. However the term "Bayesian Optimization" is very specific and does not refer to the general combination of Bayesian methods and optimization. It refers to the special case where the objective function itself is seen as random and endowed with a prior distribution, updated along evaluations (See Shahriari et al. 2016 for a review). Kriging can be referred to as Bayesian inference. Approaches classified in both optimization or probabilistic methods may then be qualified as Bayesian. This is why we think it would not particularly help to compare our method specifically to other Bayesian approaches out of the scope of Bayesian optimization. We will update the introduction to clarify the meaning of the term 'Bayesian' and to explain why a comparison with other Bayesian approaches would not make much sense.

In Butera et al. [2013] (who developed the SRSI approach, a "stochastic procedure which finds the source location and the release history by means of a Baseyian geostatistical approach" as stated by Cupola et al. [2015]), kriging is used to infer release functions that are combined with specific zonal transfer functions. Flow and transport numerical simulations need to be computed for all candidate locations of the transfer functions. In our approach, computations are not limited to isolated locations, and it is possible to explore solution at any location of the search grid (which is limited by the resolution of the flow and transport numerical model).

The approach proposed by Zeng et al. [2012] uses a Bayesian formulation of the inverse problem to estimate the probability density function (pdf) of the location and release time of a contaminant. Optimization approaches, such as ours, provide optimal solutions but it do not allow to estimate the pdf.

The paper by Raziyeh Farmani et al.[2009] does not seem relevant for comparison, as it does not deal with contaminant source identification, but with water management.
2, Why do you choose source A (89, -36) and B (100, 10) as reference contaminant sources? Besides, it is better to show source A and B in Figure 1.

We chose two different contaminant source reference locations to test the influence of the source location versus the geology first on the misfit objective function and second on the ability of the proposed approach to deal with more or less complex objective functions. This is an argument we will add in the related paragraph in Section 2 (Synthetic test cases).

3,On page 10, can you explain why there is large departure between reference source location and minimum of the objective function location.

This is because the resolution of the search grid is not the same as the grid resolution of the flow and transport numerical model. It is explained in lines 13 to 15 of page 10 and this choice is motivated between line 23 of page 5 and line 3 of page 6.

**Specific comments:**

1, On page 1, it is better to briefly explain how analytical solution and regression approaches works, just as you have done to the other three categories.

Certainly, we could precise: 'in which a set of equations can be solved analytically or whose parameters can be estimated by least-square regression'.

2, Please be careful to use "To the best of our knowledge" on page 2 and 3.

This is what we use on page 2 lines 27 and 32, we will correct this on page 3 line 12.

3, It is not appropriate to set longitudinal dispersivity as 1 m when your resolution of the aquifer is 1m*1m.

We do not agree for two reasons. First, we are mainly modeling the spreading of the contaminant due to an explicit description of geological heterogeneity at a small scale. Therefore, the longitudinal dispersivity is taken at the smallest possible value with our mesh size. Second, the same parameters are used for all the simulations. The only

unknown is the position of the contaminant source. Therefore, the comparison between the various numerical simulations is fair. To conclude, we could use a larger dispersity but it would spread the solute in an exaggerate manner and mask the effect of the heterogeneity. The problem would be much less interesting.

4, Please show both source A and B in Figure 3 and Figure 1.

This is something we can add if the referees think it can improve the understanding of the experimental setup. In that case, for consistency between the figures, it might make sense to add the search zone as well, and to add both search zone and reference locations on Figure 2 as well.

5, The dots in Figure 5 are with three different colors, and you just explained the meaning of blue and white dots. Please explain the left one.

Indeed, a part of the legend is missing. The following legend (Fig. 1 of this answer) will be added to the figure.

*References used in the answer:*

Shahriari, Bobak, Kevin Swersky, Ziyu Wang, Ryan P. Adams, and Nando de Freitas. "Taking the human out of the loop: A review of bayesian optimization." *Proceedings of the IEEE* 104, no. 1 (2016): 148-175.

Cupola, Fausto, Maria Giovanna Tanda, and Andrea Zanini. "Laboratory sandbox validation of pollutant source location methods." *Stochastic environmental research and risk assessment* 29, no. 1 (2015): 169-182.

Zeng, Lingzao, Liangsheng Shi, Dongxiao Zhang, and Laosheng Wu. "A sparse grid based Bayesian method for contaminant source identification." *Advances in water resources* 37 (2012): 1-9.

Farmani, Raziyeh, Hans Jørgen Henriksen, and Dragan Savic. "An evolutionary Bayesian belief network methodology for optimum management of groundwater contamination." *Environmental Modelling & Software* 24, no. 3 (2009): 303-310.

Butera, Ilaria, Maria Giovanna Tanda, and Andrea Zanini. "Simultaneous identification of the pollutant release history and the source location in groundwater by means of a geostatistical approach." *Stochastic Environmental Research and Risk Assessment* 27, no. 5 (2013): 1269-1280.

[Figure]

○ 9 points initial design     ● 41 first iterations     ● 50 last iterations

✕ contaminant reference location     ☐ objective function minimum

**Fig. 1.** missing legend

---

## Referee Comment (RC2) · Anonymous Referee #2 · 29 Sep 2017

This paper addressed an important problem for groundwater protection: contaminant source localization, and the authors proposed that this problem can be transferred into an optimization problem with highly non-linear objective functions, furthermore, they can use Bayesian global optimization approach to find the minimum of the objective function then thus localize the polluter. The authors then implemented the proposed optimization method into two realistic 2D synthetic cases to test and demonstrate its efficiency.

I do think the idea of this paper which uses global optimization method for finding contaminant source is interesting. However, this paper needs to be improved, and more work are needed, as listed below.

Comments:

[Figure]

1. The grammar of this manuscript needs some polish, there are some grammar errors and weird expressions in the manuscript (e.g., the usage of "firstly", "secondly" in the second paragraph of Page 3).

2. The novelty and objectives of this research need to be improved, clarified and emphasized. Although the authors used test cases with more "realistic" hydrogeological property comparing with the previous works, they are still synthetic (no fundamental difference in my opinion). And I believe the test of efficiency for a long existing optimization algorithm is unnecessary. The necessity and novelty of this work need to be clarified.

3. I am not sure the differences between two synthetic test cases, why did the authors use two very similar synthetic cases?

4. The second paragraph of Introduction on Page 1: is it necessary to introduce the other three methodologies of groundwater pollution source identification? Why did the authors bring them up in the third paragraph of Page 2? And what are the relationships between them and optimization method?

5. Line 9 in Page 2: the meaning of term "latter" is unclear, do the authors mean the second sub-class or the third one?

6. Line 33 in Page 2: please define "realistic" and explain why the previous work are not "realistic" but this study is "realistic".

7. Line 10 in Page 5: please explain why and how did the authors use the multi-Gaussian distributed initial contaminant mass.

---

## Author Comment (AC2) · 13 Oct 2017

The original comments of referee 2 are in black and our answers to these comments are in blue.

**General comments:**

1. The grammar of this manuscript needs some polish, there are some grammar errors and weird expressions in the manuscript (e.g., the usage of "firstly", "secondly" in the second paragraph of Page 3).

We carefully proofread our manuscript, but English being not the mother language of the authors, some grammatical errors might persist. We will let our manuscript proofread again by a native English speaker.

[Figure]

2. The novelty and objectives of this research need to be improved, clarified and emphasized. Although the authors used test cases with more "realistic" hydrogeological property comparing with the previous works, they are still synthetic (no fundamental difference in my opinion). And I believe the test of efficiency for a long existing optimization algorithm is unnecessary. The necessity and novelty of this work need to be clarified.

The novelty and objectives of this research are stated in the introduction (page 3, lines 14 to 22). We can add some clarifications in the introduction if necessary.

3. I am not sure the differences between two synthetic test cases, why did the authors use two very similar synthetic cases?

We propose two synthetic cases because different geological settings can lead to very different objective functions and it is important to test the robustness of the optimization method. We will add this explanation in the description of the synthetic test cases. Though the geological maps are obtained from the same training image, they are pixelwise very different, which implies different flow-paths, specific contaminant transport and results in objective functions with different structures of local minima.

4. The second paragraph of Introduction on Page 1: is it necessary to introduce the other three methodologies of groundwater pollution source identification? Why did the authors bring them up in the third paragraph of Page 2? And what are the relationships between them and optimization method?

We are simply reviewing previous work to clarify the relations between our work and previous research. We believe that this is useful for readers who may not be familiar with the details of this topic.

**Specific comments:**

5. Line 9 in Page 2: the meaning of term "latter" is unclear, do the authors mean the second sub-class or the third one?

We mean the last (third) sub-class and we will clarify it in the manuscript.

6. Line 33 in Page 2: please define "realistic" and explain why the previous work are not "realistic" but this study is "realistic".

We wrote "geologically realistic medium", so we mean realistic from a geological point of view, in contrast to constant parameter fields or realizations of stationary Gaussian random fields. In the definition of our objectives (page 3, lines 14 to 22), we also made it clear what we mean by the term realistic: "property contrasts" and "connected structures".

7. Line 10 in Page 5: please explain why and how did the authors use the multi-Gaussian distributed initial contaminant mass.

The initial contaminant mass distribution is chosen as following a multi-Gaussian distribution as a simple way to model surface spills that usually present some diffusion characteristics in their shape and can cover different geological features. We will clarify this in the manuscript.

---

## Author Response (AR1)

Dear Editor and reviewers,

Here is our answer letter, joined with our revised manuscript. The Editor comments are in cyan. Reviewer 1 comments are recalled in red, and Reviewer 2 comments are in blue. Our answers are in black, and, **in bold font, the page and line number we indicate in our answers, refer to the manuscript with highlighted changes**.

**Editor comments**

**Editor Decision: Reconsider after major revisions (further review by editor and referees) (16 Nov 2017) by Bill Hu**
Comments to the Author:
I received comments from two reviewers, both suggested "major revision". Both reviewers pointed out the authors did not clearly present the novelty of their manuscript, compared with previous studies. Both reviewers think there are some issues in the study on the two synthetical cases are not clearly presented. The writing also needs polish.
After receiving the comments, I carefully read the manuscript again, and concur with the reviewers. The authors have also provide their responses to the comments and questions reasonably. Please revise the manuscript according to the comments made by the two reviewers. The revised manuscript will be reviewed again by the two reviewers.
We Revised the manuscript accordingly to the reviewer comments and following the responses we gave online. The manuscript was carefully proofread by a native English speaker. Below, we recall the comments of the reviewers as well as our answers. In addition, we precise which pages and lines of the highlighted manuscript are affected by the related updates.

**Reviewer 1 comments**

**General comment:**

1. In the introduction section, you have introduced some classification for groundwater pollution source identification. In terms of the Bayesian approach you used, you also need to introduce the state of art of contaminant source localization based on Bayesian approaches (eg., Cupola et al. [2015], Zeng et al. [2012] , Raziyeh Farmani et al.[2009]), and detail the difference and merit of the approach you used by comparing with those Bayesian-based approaches.

   The term 'Bayesian' is typically used when a method involves Baye's rule. And, as the referee points out, many Bayesian approaches have been used throughout groundwater sciences and notably for contaminant source localization. However the term "Bayesian Optimization" is very specific and does not refer to the general combination of Bayesian methods and optimization. It refers to the special case where the objective function itself is seen as random and endowed with a prior distribution, updated along evaluations (See Shahriari et al. 2016 for a review). Kriging can be referred to as Bayesian inference. Approaches classified in both optimization or probabilistic methods may then be qualified as Bayesian. This is why we think it would not particularly help to compare our method specifically to other Bayesian approaches out of the scope of Bayesian optimization. We updated the introduction to clarify the meaning of the term 'Bayesian' and to explain why a comparison with different Bayesian approaches would not make sense **(see page 2, lines 17 to 22)**.

   In Butera et al. [2013] (who developed the SRSI approach, a "stochastic procedure which finds the source location and the release history by means of a Baseyian geostatistical approach" as stated by Cupola et al. [2015]), kriging is used to infer release functions that are combined

with specific zonal transfer functions. Flow and transport numerical simulations need to be computed for all candidate locations of the transfer functions. In our approach, computations are not limited to isolated locations, and it is possible to explore solution at any location of the search grid (which is limited by the resolution of the flow and transport numerical model).

The approach proposed by Zeng et al. [2012] uses a Bayesian formulation of the inverse problem to estimate the probability density function (pdf) of the location and release time of a contaminant. Optimization approaches, such as ours, provide optimal solutions but it do not allow to estimate the pdf.

The paper by Raziyeh Farmani et al.[2009] does not seem relevant for comparison, as it does not deal with contaminant source identification, but with water management.

2. Why do you choose source A (89, -36) and B (100, 10) as reference contaminant sources? Besides, it is better to show source A and B in Figure 1.

We chose two different contaminant source reference locations to test the influence of the source location versus the geology first on the misfit objective function and second on the ability of the proposed approach to deal with more or less complex objective functions. This is an argument we added in the related paragraph in Section 2 (Synthetic test cases **see page 4, lines 28 to 29**).

3. On page 10, can you explain why there is large departure between reference source location and minimum of the objective function location.

This is because the resolution of the search grid is not the same as the grid resolution of the flow and transport numerical model. It is explained in **lines 11 to 14 of page 11** and this choice is motivated between **lines 7 to 18 of page 8**.

**Specific comments:**

4. On page 1, it is better to briefly explain how analytical solution and regression approaches works, just as you have done to the other three categories.

Certainly, we precise: 'in which a set of equations can be solved analytically or whose parameters can be estimated by least-square regression' **(see page 2, line 5)**.

5. Please be careful to use "To the best of our knowledge" on page 2 and 3.

This is what we use on **page 3 lines 7 and 11**, we corrected this **page 3 line 28**.

6. It is not appropriate to set longitudinal dispersivity as 1 m when your resolution of the aquifer is 1m*1m.

We do not agree for two reasons. First, we are mainly modeling the spreading of the contaminant due to an explicit description of geological heterogeneity at a small scale. Therefore, the longitudinal dispersivity is taken at the smallest possible value with our mesh size **(see page 5, line 1+)**. Second, the same parameters are used for all the simulations. The only unknown is the position of the contaminant source. Therefore, the comparison between the various numerical simulations is fair. To conclude, we could use a larger dispersity but it would spread the solute in an exaggerate manner and mask the effect of the heterogeneity. The problem would be much less interesting.

7. Please show both source A and B in Figure 3 and Figure 1.

Figure 3 gives an example of Misfit objective function calculated when the true source is at A(89,-36). The cross + sign indicates the location of A. (We did not specify A(89,-36) in the

caption, so thank you for pointing this out). But because this figure is for the case when the true source is A, we did not show the source B in the figure.

This is something we added. In that case, for consistency between the figures, it makes sense to add the search zone as well, and to add both search zone and reference locations on Figure 2 as well.

8. The dots in Figure 5 are with three different colors, and you just explained the meaning of blue and white dots. Please explain the left one.

Indeed, a part of the legend is missing. The following legend was added to the figure.

○ 9 points initial design      ● 41 first iterations      ● 50 last iterations

✕ contaminant reference location      ☐ objective function minimum

*References used in the answer:*
Shahriari, Bobak, Kevin Swersky, Ziyu Wang, Ryan P. Adams, and Nando de Freitas. "Taking the human out of the loop: A review of bayesian optimization." *Proceedings of the IEEE* 104, no. 1 (2016): 148-175.
Cupola, Fausto, Maria Giovanna Tanda, and Andrea Zanini. "Laboratory sandbox validation of pollutant source location methods." *Stochastic environmental research and risk assessment* 29, no. 1 (2015): 169-182.
Zeng, Lingzao, Liangsheng Shi, Dongxiao Zhang, and Laosheng Wu. "A sparse grid based Bayesian method for contaminant source identification." *Advances in water resources* 37 (2012): 1-9.
Farmani, Raziyeh, Hans Jørgen Henriksen, and Dragan Savic. "An evolutionary Bayesian belief network methodology for optimum management of groundwater contamination." *Environmental Modelling & Software* 24, no. 3 (2009): 303-310.
Butera, Ilaria, Maria Giovanna Tanda, and Andrea Zanini. "Simultaneous identification of the pollutant release history and the source location in groundwater by means of a geostatistical approach." *Stochastic Environmental Research and Risk Assessment* 27, no. 5 (2013): 1269-1280.

**Reviewer 2 comments**

**General comments:**

9. The grammar of this manuscript needs some polish, there are some grammar errors and weird expressions in the manuscript (e.g., the usage of "firstly", "secondly" in the second paragraph of Page 3).

We carefully proofread our manuscript, but English being not the mother language of the authors, some grammatical errors might persist. The manuscript has been proofread again by a native English speaker.

10. The novelty and objectives of this research need to be improved, clarified and em- phasized. Although the authors used test cases with more "realistic" hydrogeological property comparing with the previous works, they are still synthetic (no fundamental difference in my opinion). And I believe the test of efficiency for a long existing opti- mization algorithm is unnecessary. The necessity and novelty of this work need to be clarified.

The novelty and objectives of this research are stated in the introduction. We added some clarifications in the abstract and in the introduction (**page 1 line 5 to 11, page 1 line 19 and 20, page 3 line 30 to page 4 line 7**).

11. I am not sure the differences between two synthetic test cases, why did the authors use two very similar synthetic cases?

We propose two synthetic cases because different geological settings can lead to very different objective functions and it is important to test the robustness of the optimization method. We will add this explanation in the description of the synthetic test cases. Though the geological maps are obtained from the same training image, they are pixel-wise very different, which implies different flow-paths, specific contaminant transport and results in objective functions with different structures of local minima **(see page 4, line 28)**.

12. The second paragraph of Introduction on Page 1: is it necessary to introduce the other three methodologies of groundwater pollution source identification? Why did the authors bring them up in the third paragraph of Page 2? And what are the relationships between them and optimization method?

We are simply reviewing previous work to clarify the relations between our work and previous research. We believe that this is useful for readers who may not be familiar with the details of this topic.

**Specific comments:**

13. Line 9 in Page 2: the meaning of term "latter" is unclear, do the authors mean the second sub-class or the third one?

We mean the last (third) sub-class and we clarified it in the manuscript **(see page 2, line 17)**.

14. Line 33 in Page 2: please define "realistic" and explain why the previous work are not "realistic" but this study is "realistic".

We wrote "geologically realistic medium", so we mean realistic from a geological point of view, in contrast to constant parameter fields or realizations of stationary Gaussian random fields. In the definition of our objectives **(see page 3, line 12, lines 31 and 32)**, we also made it clear what we mean by the term realistic: "property contrasts" and "connected structures".

15. Line 10 in Page 5: please explain why and how did the authors use the multi-Gaussian distributed initial contaminant mass.

The initial contaminant mass distribution is chosen as following a multi-Gaussian distribution as a simple way to model surface spills that 
[revised manuscript text omitted]

---

## Referee Report (RR1)

Dear Editor and authors,

This work can be interesting. It aims to localize the contaminant source of (non-reactive) solute in a synthetic confined aquifer through a Bayesian optimization approach, with the support of 25 measurements locations. The measurements (with zero measurement error) are taken from the 'true' case where the model inputs are assumed to be completely known. To what I have studied, authors wanted to use four scenarios (coping with different nonlinearity level of objective functions) to explore the capability of the Bayesian optimizations in localizing contaminant sources. However, in my opinion, using the objective functions stem from those four scenarios to convince the reader that these objective functions can be referred as a benchmark is weak. I notice that the authors' response to General Comments (3) of referee #2 do make sense in some extent. However, the associated nonlinearity level in these objective functions can hardly be previously classified (or say ranked) which weaken attractiveness of this work. To improve the quality of this work, I would like to suggest the authors to further perform scenarios considering measurement error in several different magnitudes and/or various measurement network having various number of measurement locations. My points to provide this suggestion are majorly attributed to that (*i*) measurement error and number of measurement locations are two very important factors in a realistic problem; (*ii*) the objective function stemmed from localizing contaminant sources is indeed a function both measurement error and number of measurement locations. Generally, the higher measurement error the higher nonlinearity level of objective functions; the less number of measurement the higher nonlinearity level of objective functions. Then, authors can explore robustness of Bayesian optimization approach in several classified nonlinearity levels, which can improve the quality of this work and increase its attractiveness of being a good reference in localizing contaminant sources.

I further give the following specific comments line by line.

Line 1 to line 2 on page 1: Please keep the consistency between terminology "transmissivity" and "hydraulic conductivity" throughout the main text.

Line 5 to line 6 on page 1: Why the objective function you proposed can be used as a benchmark? Beside they own multiple local minmia, are there other special reasons? There are many studies contributed to localize contaminant sources in heterogeneity medium. It can be better if authors can explain this in Abstract.

Line 9 to line 10 on page 4: Why did authors use 100 replications? Please explain this.

Figure 1: There are only three transmissivity values (i.e. 1E-1, 1E-3 and 1E-5) in Fig. 1, right? If so, I would like to suggest the authors to replace the color bar with a legend in color box to avoid confusion.

The caption of Figure 2: Where are the boundary conditions? Please check Figure 2.

About section 4, Can you further descript the implementation of the optimization procedure in a Flow Chart? It would be better to show computational procedure in a Figure than solely in a lot of words.

Line 20 to line 21 on page 9: Please write the preset maximum number of iteration here.

---

## Referee Report (RR2)

Dear Editor and authors,

According to the authors' revision, I make my comments as the following.

General comments
This work, at least, in its current form is unacceptable. I believe that three main points are needed to be solved to consider the manuscript for publication. First, a considerable English improvement should be made. There are many language issues which make the manuscript hard to be reviewed. Second, a deep restructure of the paper should be considered, particularly for introduction. The motivation and novelty of the paper are still unclear. In introduction, the authors use a lot of words to present the classifications of methods to identify contaminant source, while only few sentences are prepared for the Bayesian global optimization approach. Moreover, it is better that authors can classify reasons why the use of Bayesian global optimization is more attractive in comparison to its alternatives in the introduction, right? Third, authors should add some comparisons of efficiency and effectiveness between different methods tested. To show the settings can be used as a benchmark, this point is important. In this case, authors can move the sensitivity analyses of optimization results to the number of measurements and the magnitude of the observation error into a supplementary. Specific comments are given below.

Specific comments
Line 2 page 1: specify that you use deterministic hydraulic conductivity fields, right?
Line 6 page 1: please be very careful of the use of 'benchmark', because only one approach is used in the paper which didn't show reader that such settings can tell the abilities and inabilities of some existed alternatives. It is unclear that such settings can be used as a benchmark or not.
Lines 4-7 page 2: please rephrase this sentence
Line 28 page 2: please check the terminology "Parameter models"
Line 4 page 2: please add references respectively after "homogeneous" and "multi-Gaussian"
Lines 3-8 page 2: be very careful of these sentences. It should be very clear that why the geological medium you use is more proper than the others, for example, the multi-Gaussian like random field? I mean you should provide more related details. In addition, I believe that the use of "realistic" may be improper.
Line 10-11 page 3: please rephrase this sentence and check the terminology "simulated measurements"
Lines 20-23 page 3: please rephrase this sentence
Lines 32-34 page 3: again, at least now, you can't say that the settings can be used as a benchmark to tell which optimization method is better, right?
Line 4: replace "model" with "aquifer" or "field of hydraulic conductivity"?
Lines 4-6: the synthetic aquifer is used to simulate the braided-river aquifer. You should declare this early, right after Line 8 page 2.
Lines 8-12: please rephrase these sentences.

Lines 17-20: check the punctuations

Introduction: can you trim the text for classifying the methods to identify contaminant source characteristics? And it would be better that more text concerning Bayesian global optimization is specified, especially, why you choose this approach?

Line 26 page 4: What is MPS? You should say it is multi-point statistics.

Lines 27-29: this sentence is unclear. What do you mean by "contaminant spreading is mainly modeled by the explicit description of geological heterogeneity"? the logic of this sentence is incorrect. I can't understand why "longitudinal dispersivity is taken as the smallest possible value with the grid size" is attributed to "contaminant spreading is mainly modeled by the explicit description of geological heterogeneity"? Additionally, what do authors mean by "the smallest possible value"? Please be clear.

Algorithm 1: please specify that $N = 100$ and $n0 = 9$, right? You also need to tell the reader what are n, n0 and N? Please be clear.

Line 10 page 8: as you now take the measurement errors into account, the minimum of this function may not equal to 0.

Line 10 page 8: "which corresponds to an $l^p$ norm." is unclear. Please rephrase it. You mean the $l^p$ norm of what? Please be clear. Furthermore, replace "an" with "the".

Line 4 page 10: K should be in bold, because it is a symbol indicating a matrix. Please check this issue throughout the manuscript.

Equation (2): What's "p"?

Line 5 page 9: please add references for "machine learning"

Line 16 page 9: please add references for "Gaussian Processes"

Lines 17-19 page 10: remove "First" and rephrase this sentence

Line 25 page 10: the $l^2$ norm of what?

Lines 8-10 page 12: please rephrase this sentence

Line 11 page 12: what do you mean by "replications"? Please check this terminology.

Lien 12 page 12: Please rephrase this sentence.

Line 15 page 12: what do you mean by "true minimum"

Lines 12-13 page 16: rephrase this sentence.

Line 16 page 16: I didn't fine where you show the results you mention. And, what do it mean by "for $l^1$ norm objective functions"?

Lines 35-37 page 17: Please rephrase this sentence

Lines 13-15 page 17: Please rephrase these sentences

Editorial comments

Line 6 page 1: replace "or" with "and"

Line 20 page 1: classified

Line 22 page 1: replace "are" with "is" and check this throughout the manuscript

Line 14 page 2: remove "as defined above"

Line 32 page 2: contains

Line 17 page 3: remove "A"

Line 3 page 7: replace "the figure" with Figure 3

Line 6 page 8: is denoted as

Line 20 page 8: contaminant source identification problem

Line 19 page 10: analyses
The first line page 12: the explorations of the objective functions
Line 5 page 12: the explored locations
Line 7 page 12: replace "&" with "and"
Line 16 page 12: Figures 6A to D?
Line 18 page 12: Figures 6E to H?

---

## Referee Report (RR3)

Dear Editor and authors,

The authors made an appreciated improvement. Many thanks. I only have few editorial comments and recommend this manuscript for publication in HESS.

**Editorial comments**

Page 1 line 13: localizes?
Page 2 lines 17-18: Please check this sentence
Page 2 line 32: remove "a"
Page 3 line 4: suggest to replace "dominated" with "constrained"
Page 4 line 31: test cases?
Page 8 lines 6-7: please check "this procedure and geometric allows"
Page 8 line 8: replace "work either" with "either work"

---

## Author Response (AR2)

Dear Editor and reviewers,

Here is our answer letter, joined with our revised manuscript. The Editor comments are in cyan. and Reviewer 3 comments are in blue, Reviewer 2 comments are recalled in red. Our answers are in black, and, **in bold font, the page and line number we indicate in our answers, refer to the manuscript with highlighted changes**.

**Editor Decision: Reconsider after major revisions (further review by editor and referees) (07 Mar 2018) by Bill X. Hu**

Comments to the Author: Please submit a revised version based on the reviews submitted on 16 and 23 January.

We Revised the manuscript accordingly to the reviewer comments. The manuscript was carefully proofread by a native English speaker. Below, we recall the comments of the reviewers as well as our answers. In addition, we have included a point-by-point response to the reviewers' comments, indicating our revisions in the text by page and line numbers.

**Anonymous Referee #3 - Submitted on 16 Jan 2018**

**General comments:** This work can be interesting. It aims to localize the contaminant source of (non-reactive) solute in a synthetic confined aquifer through a Bayesian optimization approach, with the support of 25 measurements locations. The measurements (with zero measurement error) are taken from the "true" case where the model inputs are assumed to be completely known. To what I have studied, authors wanted to use four scenarios (coping with different nonlinearity level of objective functions) to explore the capability of the Bayesian optimizations in localizing contaminant sources. However, in my opinion, using the objective functions stem from those four scenarios to convince the reader that these objective functions can be referred as a benchmark is weak. I notice that the authors' response to General Comments (3) of referee #2 do make sense in some extent. However, the associated nonlinearity level in these objective functions can hardly be previously classified (or say ranked) which weaken attractiveness of this work. To improve the quality of this work, I would like to suggest the authors to further perform scenarios considering measurement error in several different magnitudes and/or various measurement network having various number of measurement locations. My points to provide this suggestion are majorly attributed to that (i) measurement error and number of measurement locations are two very important factors in a realistic problem; (ii) the objective function stemmed from localizing contaminant sources is indeed a function both measurement error and number of measurement locations. Generally, the higher measurement error the higher nonlinearity level of objective functions; the less number of measurement the higher nonlinearity level of objective functions. Then, authors can explore robustness of Bayesian optimization approach in several classified nonlinearity levels, which can improve the quality of this work and increase its attractiveness of being a good reference in localizing contaminant sources. I further give the following specific comments line by line.

We thank the reviewer for bringing up this important point about the nonlinearity level in these objective functions and we totally agree with the reviewer. This point has been discussed and added throughout the revised manuscript marked in blue. In particular, on Page 7, we have included the noise to the objective function and define $c_{obs}$ (Eq. 1), and subsequently this measurement has been used as our observed concentrations in Eq. 2.

Following the reviewer's suggestions, in Section 5.2, we have done some sensitivity analysis of the well configurations as well as the level of noises on the objective function. We showed that while the optimality gap can be improved with the number of wells, the algorithm performances were not significantly affected by the noises added.

The code is also updated in the repository (https://github.com/gpirot/BGICLP) so that the user can customize the measurement errors to their wishes.

A theoretical discussion of general Gaussian measurement errors along with the derivation of the resulting objective function covariance matrix are given in Appendix C.

**Specific comments:**

1. Line 1 to line 2 on page 1: Please keep the consistency between terminology "transmissivity" and "hydraulic conductivity" throughout the main text.

   The text has been corrected to mention only "hydraulic conductivity".

2. Line 5 to line 6 on page 1: Why the objective function you proposed can be used as a benchmark? Beside they own multiple local minmia, are there other special reasons? There are many studies contributed to localize contaminant sources in heterogeneity medium. It can be better if authors can explain this in Abstract.

   The data and script used to generate objective functions are shared as a benchmark because the functions present multiple local minima and are inspired from a practical field application. Sharing these complex objective functions provides a benchmark for global optimization techniques and should help designing new and efficient methods to solve this type of problems.

   It is explained in the abstract (**Page 1, lines 6 to 9**).

3. Line 9 to line 10 on page 4: Why did authors use 100 replications? Please explain this.

   This is due to the fact that the performance of the GP-based optimization algorithm can be significantly affected by the locations of the initial design $X_{n_0}$. Therefore, it is common to repeat the experiments using different initial designs to show robustness and reliability of the algorithm (**see Page 12 line 15**).

4. Figure 1: There are only three transmissivity values (i.e. 1E-1, 1E-3 and 1E-5) in Fig. 1, right? If so, I would like to suggest the authors to replace the color bar with a legend in color box to avoid confusion.

   The colorbar is now a color box with 3 values only.

5. The caption of Figure 2: Where are the boundary conditions? Please check Figure 2.

   Caption and related text have been updated.

6. About section 4, Can you further descript the implementation of the optimization procedure in a Flow Chart? It would be better to show computational procedure in a Figure than solely in a lot of words.

   An overview of the algorithm has been added (**see Page 9 line 9**).

7. Line 20 to line 21 on page 9: Please write the preset maximum number of iteration here.

   It has been indicated (**see Page 10 line 23**).

**Anonymous Referee #2 - Submitted on 23 Jan 2018**

**General comment:** The response and revision of the manuscript is really not satisfying.

In our first revision, we modified the manuscript on the basis of our responses to initial online comments, which seemed to be reasonable according to the Editor. In this second revision, we reconsidered each point individually and re-updated our manuscript when needed. We recall these comments below and give some more information about the corrections.

Please note that this time we disagree with the reviewer's comment because he does not justify what is not satisfying, according to him, in our revision. Therefore, it is not possible to guess what requires to be corrected more deeply and we cannot account for such a vague comment more than what we did so far.

**General comments from previous review:**

1. The grammar of this manuscript needs some polish, there are some grammar errors and weird expressions in the manuscript (e.g., the usage of "firstly", "secondly" in the second paragraph of Page 3).

   We carefully proofread our manuscript, but English being not the mother language of the authors, some grammatical errors might persist. The manuscript has been proofread again by a native English speaker.

2. The novelty and objectives of this research need to be improved, clarified and em- phasized. Although the authors used test cases with more "realistic" hydrogeological property comparing with the previous works, they are still synthetic (no fundamental difference in my opinion). And I believe the test of efficiency for a long existing opti- mization algorithm is unnecessary. The necessity and novelty of this work need to be clarified.

   The novelty and objectives of this research are stated in the introduction. We added some clarifications in the abstract and in the introduction **(page 1 line 5 to 10, page 1 line 18 and 19, page 3 line 26 to 35)**.

3. I am not sure the differences between two synthetic test cases, why did the authors use two very similar synthetic cases?

   We propose two synthetic cases because different geological settings can lead to very different objective functions and it is important to test the robustness of the optimization method. We will add this explanation in the description of the synthetic test cases. Though the geological maps are obtained from the same training image, they are pixel-wise very different, which implies different flow-paths, specific contaminant transport and results in objective functions with different structures of local minima **(see page 4, line 23)**.

4. The second paragraph of Introduction on Page 1: is it necessary to introduce the other three methodologies of groundwater pollution source identification? Why did the authors bring them up in the third paragraph of Page 2? And what are the relationships between them and optimization method?

   We are simply reviewing previous work to clarify the relations between our work and previous research. We believe that this is useful for readers who may not be familiar with the details of this topic.

**Specific comments from previous review:**

5. Line 9 in Page 2: the meaning of term "latter" is unclear, do the authors mean the second sub-class or the third one?

   We mean the last (third) sub-class and we clarified it in the manuscript **(see page 2, line 15)**.

6. Line 33 in Page 2: please define "realistic" and explain why the previous work are not "realistic" but this study is "realistic".

   We wrote "geologically realistic medium", so we mean realistic from a geological point of view, in contrast to constant parameter fields or realizations of stationary Gaussian random fields. In the definition of our objectives **(see page 3, lines 9 and 27)**, we also made it clear what we mean by the term realistic: "property contrasts" and "connected structures".

7. Line 10 in Page 5: please explain why and how did the authors use the multi-Gaussian distributed initial contaminant mass.

   The initial contaminant mass distribution is chosen as following a multi-Gaussian distribution as a simple way to model surface spills that 
[revised manuscript text omitted]

---

## Author Response (AR3)

*Dear Editor and Reviewer,*

*We thank you for your comments and for giving us the opportunity to greatly improve our manuscript. Below are our answers on how the different comments were addressed (in blue and italic). As the Editor comments present a summarized version of Referee 3 comments, they are not repeated below.*

General comments of Referee 3
This work, at least, in its current form is unacceptable. I believe that three main points are needed to be solved to consider the manuscript for publication.

**First, a considerable English improvement should be made**. There are many language issues which make the manuscript hard to be reviewed.
*We have done our best to address that. In particular, the manuscript has been proofread by native English speakers at each revision stage: Emily Voytek (Major revisions 1 & 3), and Andrew Greenwood (Major revision 2).*

**Second, a deep restructure of the paper should be considered, particularly for introduction**. The motivation and novelty of the paper are still unclear. In introduction, the authors use a lot of words to present the classifications of methods to identify contaminant source, while only few sentences are prepared for the Bayesian global optimization approach. Moreover, it is better that authors can classify reasons why the use of Bayesian global optimization is more attractive in comparison to its alternatives in the introduction, right?
*Thanks to the comments of referee 3, we have deeply revised the Abstract and Introduction (by entirely rewritting parts of them) with the aims to clarify the motivations, novelties and objectives of the paper. The presentation of methods to identify contaminant source and optimization approaches is more balanced. Strong arguments to use Bayesian optimization methods are given. Yet, the overall structure of the paper was preserved, as it clearly identifies respective parts where the synthetic problem is described, where the Bayesian optimization algorithm is explained, and where the results are presented.*

**Third, authors should add some comparisons of efficiency and effectiveness** between different methods tested. To show the settings can be used as a benchmark, this point is important. In this case, authors can move the sensitivity analyses of optimization results to the number of measurements and the magnitude of the observation error into a supplementary.
*As explained in the newly formulated introduction, we opted for a discretized version of the problem for several reasons touching notably upon open data and repeatability (yet, users have the possibility to use re-interpolated versions of the data set so as to use it them with continuous optimizers, e.g. for benchmarking global optimization algorithms). In the present discrete case, there are not really many competitors available. In a previous phase, we made some comparisons with gradient-based methods relying on re-interpolated objective functions that highlighted the superior performance of Bayesian optimization. But is was for granted as such descent algorithms are intrinsically local; besides, this created a number of complications in terms of fair comparison as it was not clear how to count gradient evaluations, not only in terms of cost but also since they relied on an approximation of f that was based on a fine grid of evaluations; we could have appealed to finite differences, but that opens new questions as well and in the end it just appeared not so relevant to go into such trouble for limited information: global beats local. Hence, and in accordance with comments from another reviewer, we favored instead exploring the robustness of Bayesian Optimization to a number of departures from the default case, ranging from discretized settings to noise within the objective function definition and also changing geology, etc. We made our best in the novel introduction to motivate and explain that clearly and in appropriate detail.*

*The sensitivity analysis, asked during Major revision 2 provides significant results that we think should be kept inside the paper.*

Specific comments

Line 2 page 1: specify that you use deterministic hydraulic conductivity fields, right?

*We do not understand the comment; by definition, a synthetic case is explicitly characterized.*

Line 6 page 1: please be very careful of the use of 'benchmark', because only one approach is used in the paper which didn't show reader that such settings can tell the abilities and inabilities of some existed alternatives. It is unclear that such settings can be used as a benchmark or not.

*Renamed 'benchmark case' following the Cambridge dictionary*

Lines 4-7 page 2: please rephrase this sentence

*The sentence has been It has been removed during rewriting of the introduction.*

Line 28 page 2: please check the terminology "Parameter models"

*It has been updated as 'Parameter sets'*

Line 4 page 3: please add references respectively after "homogeneous" and "multi-Gaussian"

*References have been added.*

Lines 3-8 page 2: be very careful of these sentences. It should be very clear that why the geological medium you use is more proper than the others, for example, the multi-Gaussian like random field? I mean you should provide more related details. In addition, I believe that the use of "realistic" may be improper.

*It has been removed*

Line 10-11 page 3: please rephrase this sentence and check the terminology "simulated measurements"

*It has been removed*

Lines 20-23 page 3: please rephrase this sentence

*The whole introduction having been reworked, these sentences have been removed.Lines 32-34 page 3: again, at least now, you can't say that the settings can be used as a benchmark to tell which optimization method is better, right?*
*'benchmark case'*

Line 4: replace "model" with "aquifer" or "field of hydraulic conductivity"?

*It has been updated*

Lines 4-6: the synthetic aquifer is used to simulate the braided-river aquifer. You should declare this early, right after Line 8 page 2.

*This comment is not clear or point to wrong locations in the paper.*

Lines 8-12 page 4: please rephrase these sentences.

*Same as the response to the point on Lines 20-23 (page 3) above.*

Lines 17-20 page 4: check the punctuations Introduction: can you trim the text for classifying the methods to identify contaminant source characteristics? And it would be better that more text concerning Bayesian global optimization is specified, especially, why you choose this approach? This point too has been accounted for within the deep changes that have been performed on the introduction.   Line 26 page 4: What is MPS? You should say it is multi-point statistics.

*It has been added*

Lines 27-29 page 4: this sentence is unclear. What do you mean by "contaminant spreading is mainly modeled by the explicit description of geological heterogeneity"? the logic of this sentence is incorrect. I can't understand why "longitudinal dispersivity is taken as the smallest possible value with the grid size" is attributed to "contaminant spreading is mainly modeled by the explicit description of geological heterogeneity"? Additionally, what do authors mean by "the smallest possible value"? Please be clear.

*The sentence has been rephrased to clarify this point.*

Algorithm 1: please specify that N = 100 and n0 = 9, right? You also need to tell the reader what are n, n0 and N? Please be clear.

*It has been added*

Line 10 page 8: as you now take the measurement errors into account, the minimum of this function may not equal to 0.

*It has been corrected.*

Line 10 page 8: "which corresponds to an l p norm." is unclear. Please rephrase it. You mean the l p norm of what? Please be clear. Furthermore, replace "an" with "the".

*We thank the referee for pointing out that more precision is desirable here. In order to remove any ambiguity, we formulated this into*
*"which corresponds to the $\ell^p$ distance between the matrices $(c_{obs}(i,t))_{i=1,\dots,25,t=1,\dots,T}$ and $(c_{sim}(\mathbf{x},i,t))_{i=1,\dots,25,t=1,\dots,T}$, where $p\geq 1$ is a parameter that can be arbitrarily chosen by the modeller (in our experiments both $p=1$ and $p=2$ were considered, as mentioned later).*

Line 4 page 10: K should be in bold, because it is a symbol indicating a matrix. Please check this issue throughout the manuscript.

*Done.*

Equation (2): What's "p"?

*See penultimate response: p is a parameter that governs the distance between reference measurements and simulation results obtained with candidate contaminant source locations.*

Line 5 page 9: please add references for "machine learning"

*→ This part of the sentence has been rephrased as "relies on a machine learning approach relying on Gaussian Process (GP) models \cite{rasmussen2006}"*

Line 16 page 9: please add references for "Gaussian Processes"

*Following up the change in response to the last point, the concerned sentence has been replace by "GPs constitute a very popular class of probabilistic models that are fully specified by a mean function $m\left(\mathbf{x}\right)$ and a covariance function $k\left(\mathbf{x},\mathbf{x}'\right)$ \cite{rasmussen2006}". Lines 17-19 page 10: remove "First" and rephrase this sentence*
*It has been updated*

Line 25 page 10: the l 2 norm of what?

*For more clarity, we have reformulated the concerned piece of sentence into "where the noise level $\kappa$ (of Eq.~\ref{eqCobsnoisy}) is set to $0$ and the parameter $p$ of the objective function $f(x)$ is set to $2$". Lines 8-10 page 12: please rephrase this sentence*
*It has been updated*

Line 11 page 12: what do you mean by "replications"? Please check this terminology.

*Replaced by 'runs'*

Lien 12 page 12: Please rephrase this sentence.

*It has been updated*

Line 15 page 12: what do you mean by "true minimum"

*The word 'true' has been It has been removed.*

Lines 12-13 page 16: rephrase this sentence.

*It has been updated*

Line 16 page 16: I didn't fine where you show the results you mention.

*This has been clarified: these results are not shown.*

Line 16 page 16, what do it mean by "for l 1 norm objective functions"?

*For the sake of clarity, we have reformulated the whole concerned part as follows: The results presented here are based on an objective function $f$ computed with $p=2$, which corresponds to an $\ell^2$ distance between reference and candidate concentration values (See*

*Eq.~\ref{eqObjFunc}). As the choice of $p$ may substantially influence the flat or deep aspect of valleys (low value zones) of the objective function, we additionally tested the EI algorithm on the 4 scenarios for objective functions with $p=1$. We found that building $f$ onto the $\ell^2$ distance leads to flatter wide valleys of low values for the objective functions, which might not favor the efficiency of the EI optimizer. However, the results and performances of the EI algorithm are very similar between the two norms tested. This is why we decided not to show the results of the algorithm objective functions built upon the $\ell^1$ distance. Lines 35-37 page 17: Please rephrase this sentence*
*It has been updated*

Lines 13-15 page 17: Please rephrase these sentences
*It has been updated*

**Editorial comments**

Line 6 page 1: replace "or" with "and"
*It has been updated*

Line 20 page 1: classified
*It has been removed*

Line 22 page 1: replace "are" with "is" and check this throughout the manuscript
*It has been updated*

Line 14 page 2: remove "as defined above"
*It has been removed*

Line 32 page 2: contains
*It has been updated*

Line 17 page 3: remove "A"
*It has been updated*

Line 3 page 7: replace "the figure" with Figure 3
*It has been updated*

Line 6 page 8: is denoted as
*It has been updated*

Line 20 page 8: contaminant source identification problem
*It has been updated*

Line 19 page 10: analyses
*No, we mean 'sensitivity analysis'*

The first line page 12: the explorations of the objective functions
*It has been updated*

Line 5 page 12: the explored locations
*It has been updated*

Line 7 page 12: replace "&" with "and"
*It has been updated*

Line 16 page 12: Figures 6A to D?
*It has been updated as 6A-D*

Line 18 page 12: Figures 6E to H?
*It has been updated as 6E-H*

[revised manuscript text omitted]